# Cohort profile: Bandar Kong prospective study of chronic non-communicable diseases

**Azim Nejatizadeh**[1]*, **Ebrahim Eftekhar**[1], **Mohammad Shekari**[1], **Hossein Farshidi**[2], **Seyed Hossein Davoodi**[3], **Mehdi Shahmoradi**[1], **Hossein Poustchi**[4], **Amin Ghanbarnejad**[5], **Teymour Aghamolaei**[5], **Hadi Yousefi**[6], **Shideh Rafati**[5]

1 Molecular Medicine Research Center, Hormozgan University of Medical Sciences, Bandar Abbas, Iran, 2 Cardiovascular Research Center, Hormozgan University of Medical Sciences, Bandar Abbas, Iran, 3 Cancer Research Center, Shahid Beheshti Medical University, Tehran, Iran, 4 Liver and Pancreatobiliary Diseases Research Center, Digestive Diseases Research Institute, Tehran University of Medical Sciences, Tehran, Iran, 5 Social Determinants in Health Promotion Research Center, Faculty of Health, Hormozgan University of Medical Sciences, Bandar Abbas, Iran, 6 Department of Nursing, Faculty of Nursing and Midwifery, Hormozgan University of Medical Sciences, Bandar Abbas, Iran

* azimnejate@yahoo.com

**Data Availability Statement:** The study is a part of the PERSIAN Cohort Study, the kind applicants should follow the respective rules and the protocol of data sharing and scientific collaboration. All the

## Abstract

Chronic non-communicable diseases (NCDs), are the leading causes of death among adults worldwide. It is projected that half of the NCDs could be avoided by preventing measures. Under the prospective epidemiological research studies in Iran (PERSIAN), we established a prospective population-based cohort study in southern Iran. The present study was designed to observe changing pattern of lifestyle transition over time and investigate the incidence and prevalence of regional modifiable risk factors as well as their associations with major NCDs. At baseline, 4063 participants aged 35–70 years were recruited on Oct, 2016and planned to get re-evaluated every 5 years along with annual follow-up. Data using validated electronic questionnaire comprising 55 questions and 482 items including general, medical and nutrition queries was collected. Blood, hair, nails, urine specimens and anthropometric measurements were taken. The response rate was 99%. In the results; male and female participants were 42.5% and 57.5%, respectively. Of note, 30.4% of women and 16.1% of men were obese. The prevalence of hypertension in men and women was 14.6% and 21%; however, diabetic men and women were 17.4% and 12.4%, respectively. Living in rural areas increased the odds of having hypertension by 1.33 (AOR = 1.33, 95% CI:1–09, 1.61). Noteworthy, logistic regression displayed that aging could predispose individuals to be more overweight, hypertensive and diabetic. The prevalence of multimorbidity of 3 or more NCDs were 8% (No. 326) and 6% (No.240), respectively. Intake of fruits, vegetables and dairy was less than two servings per day in 9.2%, 13% and 58.3% of the participants. Lower cardiovascular diseases and serum level of FBS and higher HDL level in sailors/fishermen compared to other job groups were significant (p-value <0.001). The second annual follow-up was completed and now at the end of the third wave. Findings of the present study signified the high prevalence of behavioral risk factors and their associations with respective NCDs. Subsequently, it is essential to keep track lifestyle variations, the modifiable risk factors and NCDs trends by prospective population-based cohort studies.

authors have special access to the data that other qualifying researchers would have upon applying to have. All relevant data are within the manuscript and its Supporting information files. All interested investigators in Iran and worldwide would have free access to the data of this study, and necessary processes are available at the Cohort website to reproduce the project, participate in collaborative research projects, and use the data. The data underlying the results presented in the study are available from http://persiancohort.com/access or by requesting the principal investigator via email: azimnejate@yahoo.com.

**Funding:** This work was supported by Hormozgan University of Medical Sciences through allocated grant and the Deputy of Technology and Research at the Ministry of Health and Medical Education (MOHME), Iran with assigned grant [No. 700/112, Date: Feb, 2016]. We are grateful to all of them for their support and financial provision.

**Competing interests:** The authors have declared that no competing interests exist.

**Abbreviations: 95% CI**, 95% Confidence Interval; **ESR**, erythrocyte sedimentation rate; **FFQ**, food frequency questionnaire; **NCDs**, non-communicable diseases; **OR**, Odds Ratio; **PCID**, PERSIAN Cohort Identification; **PERSIAN**, Prospective Epidemiological Research Studies in IrAN; **QA**, Quality assurance; **QC**, Quality control; **SES**, socioeconomic status; **WHO**, World Health Organization.

# Introduction

Non-communicable diseases (NCDs), known as chronic diseases, resulting from a complex interaction of genetic, behavioral and environmental factors, being defined as diseases of long duration and slow development that have led to experience high morbidity and mortality, particularly, in low and middle income countries, account for almost more than 70% of all burden of diseases worldwide. The main force drivers to NCDs are rapid urbanization, industrialization, globalization, aging and mostly originate from low physical activity, morbid diet, smoking and harmful use of alcohol consumption leading to raised blood lipids and glucose, hypertension and excess body weight. Although, NCDs affect all age groups, adults and elderly often experience the common chronic diseases, with nearly equal distribution among men and women [1, 2]. According to WHO report, NCDs are responsible for 76.4% of all deaths so as cardiovascular diseases, cancers, respiratory disorders and diabetes mellitus, each account for 45.7%, 13.5%, 3.8% 2.2% of death, respectively. With respect to the non-modifiable risk factors including aging, gender, family history and ethnicity, the major culprit of morbidity and mortality being attributed to modifiable behavioral risk factors [3]. The "premature" death due to NCDs largely happen, between ages 30 to 69 years, in low- and middle-income countries [3]. Early detection, screening and timely management, are vital components to prevent and control NCDs [3, 4]. It is expected that more than half of NCDs burden could be evaded by averting the major modifiable risk factors. The World Health Organization's Eastern Mediterranean region (EMR), has now declared that in Iran, NCDs is increasing and has got a critical condition leading to annual 2.2 million deaths and growing over time, emerging as a hot-spot of NCDs [5].

In the middle east, Iran with 1% of world's population, has also encountered unhealthy behaviors and subsequently growing NCDs where overweight, obesity and cardiovascular diseases are on the rise [6]. The result of NCDs surveillance in Iran revealed that overweight/obesity, and hypertension as the main risk factors for NCDs, are the same as the world. Approximately 800 to 850 deaths daily happen, 89% of which are mainly due to Cardiovascular (46%), cancer (14%), respiratory (6%), perinatal diseases (6%) and accidents (17%) [7, 8]. The NCDs burden and mortality rate have been distributed unevenly at national level; therefore, mapping of regional NCDs according to climate and geographic areas is of importance. The latest study in southern Iran which was conducted in Hormozgan province revealed that 25% of the adult population was at risk of NCDs. This investigation also showed higher BMI (24.3%) in women, 17.2% with hypertension, 14% with consumption of less than 5 servings of fruits and vegetables (lower than national level), 28.6% with physical inactivity (higher than national level) and at least 25% of the subjects simultaneously carried 3 main risk factors contributing to NCDs [9]. Another study in the same province showed that the rate of cigarette smoking and Hookah was above 15% around 13%, respectively, which were significantly correlated with age and education level [10]. It is well recognized that adequate region-specific evidence to NCDs is required in order to prevent and control the modifiable risk factors and then manage the disease. Undoubtedly, implementation of prospective population-based cohort studies dedicated to local cultures and ethnicities could generate region-specific evidence to evaluate a complex network of health determinants in order to design and administer unique interventions tackling NCDs [11–13]. The north of Hormozgan province is mountainous, however, to the south, it is limited to the coastal plains. Environmental exposures and life style of the coastal residents are quite different compared to mountainous areas that itself could likely lead to the development of different region-specific risk factors [14, 15]. At a glance, some measured biochemical variables in PERSIAN cohort study were presented [16–20] (Table 1). Bandar Kong cohort study as

**Table 1. Biochemical variables in various studies of PERSIAN cohort study.**

| Variables | Cohort | Hormozgan cohort | Fasa PERSIAN cohort (12) | Rafsanjan Cohort (13) | Great Khorasan Cohort (14) | PERSIAN Guilan cohort (15) | Azar cohort (16) |
|---|---|---|---|---|---|---|---|
| Sample size | | 4040 | 9975 | 9990 | 2233 | 950 | 13099 |
| FBS (mg/dL) | | 108.13±42.968 | 92.375±28.615 | 113.27±39.11 | 88.28±25.22 | 102.59±31.7 | 98.34±32.40 |
| SGOT (AST) (U/L) | | 21.95±11.129 | 22.7±8.655 | 19.87±11.81 | - | 20.08±9.8 | - |
| SGPT (ALT) (U/L) | | 27.17±21.4 | 25.19±14.27 | 21.55±15.37 | - | 21.68±16.4 | - |
| Triglycerides (TG) (mg/dL) | | 136.59±87.547 | 132.24±82.81 | 168.88±109.22 | 132.74±97.34 | 172.83±94.5 | 149.12±84.25 |
| GGT (U/L) | | 28.946±32.302 | 23.12±21.135 | - | | 27.69±23.3 | - |
| HDL-C (mg/dL) | | 47.952±10.7788 | 50.73±15.42 | 57.90±12.45 | 33.20±13.51 | 45.24±9.4 | 45.42±10.09 |
| LDL-C (mg/dL) | | 127.56±34.574 | 107.505±32.435 | 108.04±30.50 | 123.55±38.61 | 121.51±30.9 | - |
| Total Cholesterol (TC) (mg/dL) | | 202.39±42.375 | 184.605±38.67 | 198.78±41.89 | 189.18±46.33 | 201.43±37±0.2 | - |

a part of PERSIAN cohort study (http://persiancohort.com), being the first study in terms of size/design and perspectives to investigate the health profile of the inhabitants.

We aimed to determine the impact of environmental differences and exposures at coastlines on inhabitants' lifestyle and to estimate the prevalence of region-specific modifiable risk factors and their associations underlying NCDs with particular focus on sailors/fishermen, in southern Iran. Moreover, we launched and adopted a large biobank storing various human specimens under standard protocols for deciphering the role of molecular/ genomic biomarkers and environmental factors and their interactions in developing NCDs.

## Materials and methods

### Ethics and informed consent

The study was approved by the Ethical Review Board of Hormozgan University of Medical Sciences. Written informed consent was obtained from each participant, upon fulfillment of inclusion/exclusion criteria and willingness to participate.

### Geographic location of the cohort

Hormozgan, Iran's wonderland, the southernmost coastline province, is one of the hot and humid regions of Iran, have got desert and semi desert climate. The province, facing Oman and UAE, has an area of 70697 square kilometers along with 1000 km of coastline [21, 22]. There is scanty precipitation year-round. According to the latest census in 2016, it possesses more than 1.7 million people, consisting 53% men and 47% women, of which, 54.7% are urban and 46.3% rural. The main food is made up of various aquatic animals. A big number of industries and infrastructures are located in the Province. Shahid Rajaee Port, Bandar Abbas refinery, Bandar Abbas power plant, Ship building industry, and a number of dams can be listed as a few. Fishing and 30% of Iran's fishery produce, hand-built of Lenj vessels, sea transport and agriculture are the other activities of the coastal areas [23]. It has 8 coastal counties. Our Study was established in Bandar Kong county, a harbor city, in the southernmost point of Iran, standing on the height of 5 meters above sea level with hot summers and mild winters (Fig 1). The county is an ancient port with a history of ancient trade which has been registered in the UNESCO. Bandar Kong, placed in the west of Hormozgan province, is known as a historical port where was a dynamic port during Achaemenes Empire. The occupation of most people is related to the sea and sailing [24].

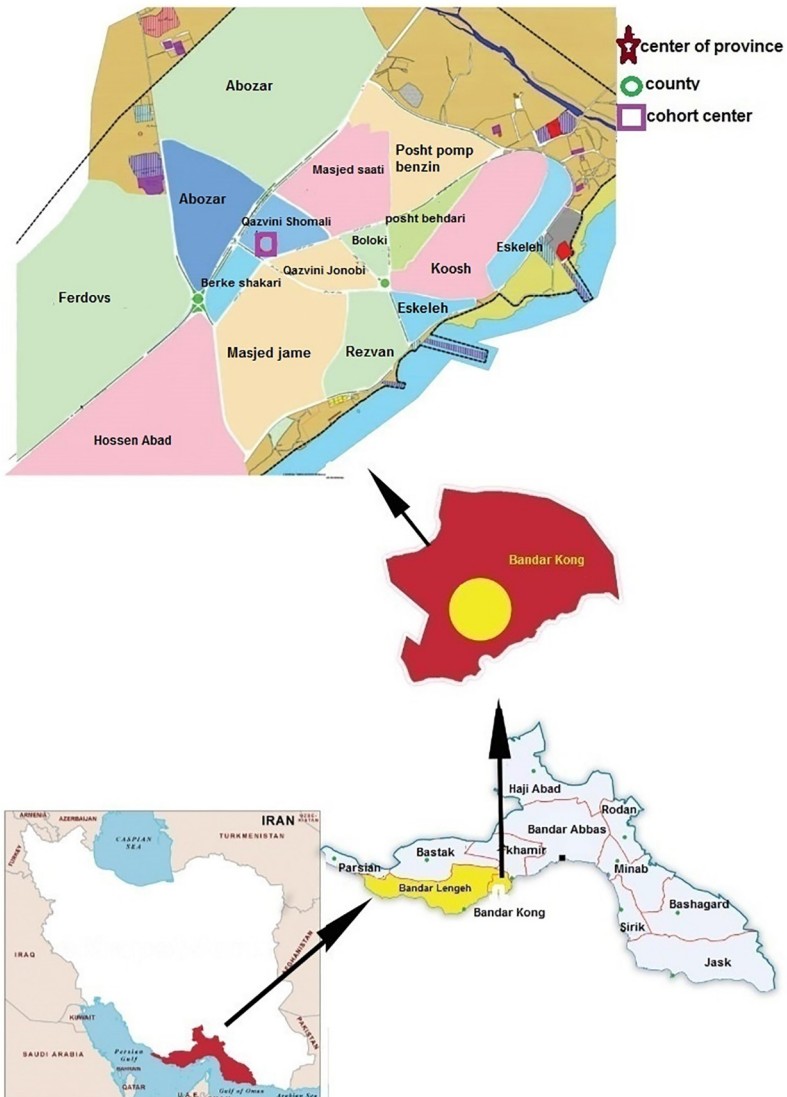

**Fig 1. The map of Iran, Hormozgan province and geographic locations of Bandar Kong cohort study in southern Iran.** Upper part of the figure approximately corresponds to the street map of cohort study field. [Republished from 1. Organization of Management and Planning of Hormozgan Province; https://www.ncc.gov.ir/images/docs/files; https://www.amar.org.ir/, under a CC BY license, with permission from the Mayor of the Bandar Kong and above-mentioned body, original copyright [2006]; 2. The image was also visualized by Google Maps (https://www.google.cat/maps/)].

## Study design

The study was designed based on PERSIAN Cohort study (S3 File). Our study was launched in October 2016 to undertake baseline survey and annual follow-ups on the health status of the inhabitants of hormozgan province in southern coastlines of the country, at least for a period of 15years. It was planned to get repeated every 5 years. The cohort center was comprised well-designed rooms for reception, enrollment, sample taking, measurements and interview coupled with an equipped laboratory with a standard biobank. All the instruments and consumable materials were procured and set up by qualified experts according to PERSIAN Cohort Protocol [25, 26].

## Study population

At first, the scientific committee enrolled 4200 participants out of 6000 permanent residents aged 35–70 years in accordance with PERSIAN Cohort arrangement. Sample collection was carried out based on the respective health center statistics and urban/rural divisions. The required sample size was calculated by Fleiss with continuity correction formula, with a two-sided significance level(1-alpha) = 0.95, power (1-beta, % chance of detecting) = 85%, the ratio of sample size of unexposed/exposed = 1, percent of unexposed with outcome = 5%, and risk/prevalence ratio or odds ratio = 1.5. So, it was estimated 2017 people for the exposed group, 2017 non-exposed group, and hence the total required sample size was estimated at 4034 people [27]. The inclusion criteria were: (i) willingness to participation, (ii) aged 35–75, (iii) residency for one year and at least nine months each year, (iv) written informed consent, (v) Iranian national. Exclusion criteria included: (i) no interest to attend, (ii) being guest, and (iii) mental or physical disabilities.

## The cohort staff

Local individuals were recruited and trained in cohort principles and interview techniques and procedures by professional researchers. The cohort staff were comprised a physician, six interviewers, a nurse, an administration support officer, a field manager, an epidemiologist, two nutritionists and a biochemist. The Study being supervised by the principal investigator and run by the executive team along with contribution of a panel of academics.

## Enrollment phase

The main phase of the study started on Oct., 27, 2016. The details were discussed by constructive meetings with local authorities, health care providers, stakeholders, city council, the mayor. We announced the benefits of the study to the public by awareness-raising activities including local media, websites, distributing brochures, pamphlets. Furthermore, a couple of meetings with local health network authorities, were held and technical details and procedures were deliberated. The health-defined block was selected from the nearest one to the primary health center. According to the assigned addresses, the interviewers met each household, explained the study and gave them a pamphlet for further details. After obtaining informed consent, each participant received a dated invitation and were reminded by phone one week and one day earlier. They were requested not to clip their nails 7–10 days before, and to keep clean the hair, as well as being fast at least for 10–12 hours overnight of the attendance day. They were invited in a designated time and location where trained staff initiated field-level data collection. A 11-digit code namely PERSIAN Cohort Identification (PCID) was assigned. Next, 25ml blood sample from each were stored in 1 to 1.5ml 2-D cryotube (Micronic, Lelystad, the Netherlands) at -80 C˚ and 15 mL of second voiding in the morning was requested. Around 500 hairs, 1–3 cm long and 1-mm nail clippings from all fingers were collected and kept in a dry and cold place. Weight (kg), height (cm), wrist, hip and waist circumferences (cm) were measured. The participants then were entertained with breakfast. The outline of the 4-step enrollment was summarized in the Fig 2.

## Data collection and assessment of exposures

Approximately 99% of the volunteers took part in the cohort study. When the participants attended the cohort center, A detailed electronic questionnaire (S1 File) including 55 questions and 482 items, divided into three key fields, viz., general, medical and nutritional components were administered by four interviewers (Table 2). Individuals who have smoked at least 100

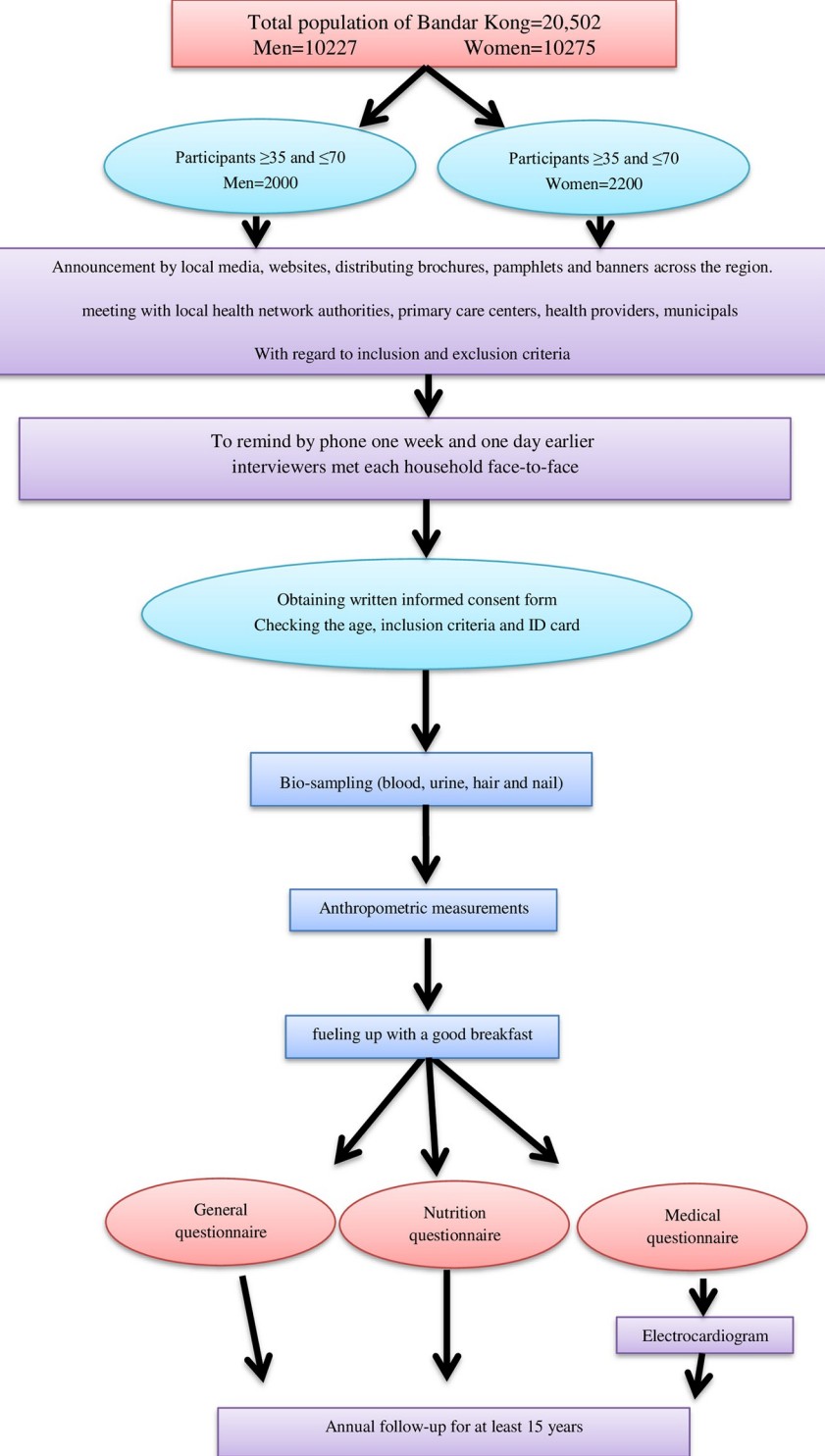

**Fig 2. The phases of the data collection process of Kong cohort study.**

**Table 2. Measures and tools administered at the enrollment phase.**

| Field | Interview | Measures |
|---|---|---|
| **Demographic and Socio-economics** | Questionnaire: 82 items | Age, sex, education, residency, employment status, ethnicity, socio-economic status, residential status, marriage status, landline and mobile phone access, fuel status at home, national and abroad journeys, |
| **Lifestyle** | Questionnaire: 69 items | Sleep quality, drug use, pesticide use, tea consumption, history of smoking, alcohol consumption, daily activity, |
| **Oral & dental health** | Questionnaire: 9 items | Tooth brushing, dental floss, mouthwash use, DMF status, Implant number/ status |
| **Physical activity** | Questionnaire: 28 items | Duration and intensity of physical activity, sport activity, daily activity at home and outside, spent hours for media |
| **Fertility History** | Questionnaire: 33 items | Menstruation, pregnancy, breastfeeding, contraceptive use |
| **History of chronic disease** | Questionnaire: 81 items | Diabetes mellitus, hypertension, cardiovascular disease, chronic lung disease, rheumatic disease, cancers, psychiatric disorders |
| **Family history of chronic disease** | Questionnaire: 27 items | history of chronic disease in the family |
| **Personal habits** | Questionnaire: 4 items | Cigarette and hookah smoking, alcohol and drug use, annual reading, |
| **Nutrition** | Questionnaire: 168 items | Food Frequency Questionnaire (FFQ), nutritional supplements, drinking water, food habits, preparation |
| **Physical examination and anthropometric** | Clinical measurements | Height, weight, waist and hip measurements, alopecia, spine disorders, Blood pressure |

cigarettes for whole life and or smoke every day or some days weekly, were considered current smoker. Regular exercise was considered as any physical activity performed at least 30 minutes for three or more days per week, except walking for work or life. Dietary pattern was measured through a food frequency questionnaire (FFQ) customized for the Iranians. A 132-item semi-quantitative FFQ was filled by trained nutritionists to assess the dietary intakes of the participants over the past year. Information of each food item was asked by two questions: 1) the frequency (number of times) of food intake per day, week, or month in the previous year, and 2) the amount of food (portion size based on the standard serving sizes commonly consumed by Iranians) commonly eaten every time. All dietary intakes were converted to grams per day by using household portion sizes of consumed foods (Table 2).

Following the questionnaire, at first, height was measured in centimeters, standardized to the nearest 0.5 cm, by means of a Height Rod Wall, FAZZINI S208, Italy, 2015. Weight was measured to the nearest 10 g, in kilograms via transportable Weighing scale, RGZ 160, China. The trained staff used SECA 201 retractable tape to measure waist and hip circumferences. Waist-hip ratio and waist-height ratio were calculated. Subsequently, physical examination was fulfilled by a trained physician. Systolic and diastolic blood pressure were measured as per the protocol. A standard mercury sphygmomanometer on the right arm twice so the first one done upon 5- minute rest in sitting position and the last one was taken 15 minutes later based on the protocols, and then the mean values were used for analysis. Pulse rate was also examined. Biochemical and hematological assays including of fasting blood glucose (mg/dl), serum cholesterol, low-density lipoprotein, high-density lipoprotein (HDL), triglycerides, urea, creatinine, uric acid, complete and differential blood count along with erythrocyte sedimentation rate (ESR) were performed. All measurements were according to a standard laboratory protocol using Pars Azmoon kits done by Hematology Cell counter (Mindray, China, 2017) and Biochemistry Auto Analyzer (BT1500, Italy, 2017). The remaining blood specimens were stored at -80C˚ until analysis in future studies (Table 3). Besides, the collected urine samples were kept in -20C˚ for macroscopic and microscopic analysis.

**Table 3. Blood taken from each individual and stored.**

| Sample type | Aliquots | Storage |
|---|---|---|
| Whole blood | Two 1.5 ml Cryo Tubes | $-80^{\circ}C$ |
| Plasma | Two 1.5 ml, and four 1 ml Cryo Tubes | $-80^{\circ}C$ |
| Buffy coat | Two 1 ml Cryo Tubes | $-80^{\circ}C$ |
| Red Blood Cells | One 1 ml Cryo Tube | $-80^{\circ}C$ |
| Serum | Two 1.5 ml Cryo Tubes | $-80^{\circ}C$ |
| **Total** | 25 ml | |

## Annual follow-ups and re-surveys

Fresh and additional detailed data being measured for the follow-up investigations to adjust and improve the data collected at the baseline. Through in-person interview via phone, this phase aimed to look for the same objectives as at enrollment phase besides: a) to evaluate demographic transitions and renew the recorded socio-demographic characteristics and addresses, b) to study lifestyle changes and respective trends, c) to collect detailed information on personal and family medical history, d) to observe and review any recent treatment, hospitalization, risk factors and NCDs or death. In case of any emerging outcome or condition, the participants are requested for further evaluations, otherwise, they will get visited in their residence. Subsequently, three medical consultants autonomously evaluate the clinical findings and assign a diagnosis code to each outcome. Hence, the two disease codes are compared, and if they are unlike, a third consultant will review and finalize the diagnosis. We, now, completed the second annual follow-up and now at the end of the third with participation rate of 99%. We designed to re-survey the whole enrollment phase including 50% of the participants every 5-year for at least 15 years along with annual follow-up.

## Quality assurance (QA) and quality control (QC)

An epidemiologist was designated as in-charge of the quality assurance and quality control. Collected data was checked daily by each interviewer and field controller. A quality control manager observed the data for missing elements, completeness and consistency using a validated checklist on a daily basis. Subsequently, a biostatiscian was committed to regular surveillance of the data quality at the weekends. Furthermore, online data monitoring and interviewers' performance were observed by central PERSIAN cohort team on a smart data server. Data clean-up being carried out monthly by local and central teams. To control and monitor the quality and accuracy of measuring devices, we calibrated all the concerned devices periodically according to the standard protocols. To prevent information bias, self-administered information was avoided and inconsistent records were corrected. Measurement methods were standardized too. Consistent measurement of inter-assay coefficients of variation was carried out just for normal levels on a regular basis (Table 4).

## Biostatistical analysis

Categorical variables using number, percent (%) and continuous variables by the mean and standard deviation (SD) have been described. The t-test was used to compare continuous variables in 2 groups. Moreover, the Chi-square test was used to examine the association between 2 categorical variables. The multivariable binary logistic regression was used to estimate the adjusted odds ratio (ORs) and determine the relationship between some of the factors with hypertension, diabetes, and heart diseases. We, here, reported the prevalence and 95%

**Table 4. Inter-assay coefficients of variation (CV) for biochemistry measurements.**

| Analyte | Level (normal/High) | CV (normal/High)[1] |
|---|---|---|
| **Fasting blood sugar** (mg/dL) | 92.2 | 3.3% |
| | 254 | 1.7% |
| **Blood Urea Nitrogen** (mg/dL) | 20.51 | 4.0% |
| | 62.6 | 4.6% |
| **Creatinine** (mg/dL) | 1.59 | 3.1% |
| | 4.32 | 3.4% |
| **Triglycerides** (mg/dL) | 97 | 2.8% |
| | 222 | 2.0% |
| **Total cholesterol** (mg/dL) | 144 | 2.0% |
| | 190 | 2.2% |
| **SGOT (U/L)** | 51.3 | 3.8% |
| | 171 | 4.8% |
| **SGPT (U/L)** | 55.8 | 3.2% |
| | 109 | 3.0% |
| **ALP (IU/L)** | 121 | 3.8% |
| | 296 | 4.9% |
| **HDL Cholesterol**(mg/dL) | 43 | 1.8% |
| **Gamma-GlutamylTransferase (GGT) (IU/ml)** | 29.7 | 4.0% |
| | 75.4 | 3.0% |

[1]The inter assay variation describes the variation of results obtained from repeated experiments. It is expressed by inter assay coefficient of variation (inter assay CV) to monitor the precision of results between different assays.

confidence intervals of major risk factors to NCDs. In present study, using statistical methods such as regression models, the relationship between independent variables and the desired outcomes in different categories of confounding variables were calculated separately or controlled by analysis stratification. All statistical tests were two-sided as well as all analyses were carried out using the SPSS software; A P < 0.05 was considered as statistically significant.

## Results

Data was cleaned up and validated. Participation rate was almost 94% and declining to participate was due to unwillingness, lack of confidence and feeling unwell. The response rate was 99%. Baseline demographic characteristics were shown in Table 5. Out of 4063 participants, 42.5% were male, 57.5 female, 84.8% were resident of urban area and the remaining, 15.2%, of rural area. The marriage rate was 89.5% and 67.8% were illiterate or with inadequate literacy. One third of the participants was in the age group of 40–49 years. All the participants were native where 44.2% of them as an employee and remaining 47.7%, as housewife. Housing ownership was 85%. Participants' baseline characteristics by gender and prevalence of self-reported health conditions have been displayed in Table 6. Of note, 35.4% of men and 37.0% of women were overweight, while, 30.4% of women were obese compared to 16.1% obesity in men. Hypertension was observed in 27.3% of the participants (27.3% (95% CI: 25.9–28.6%) of whom the highest proportion (64.7%) was found in age above 65 years. The mean systolic and diastolic BP were 118.7 ± 17.5 (SD) mmHg and 76.9 ± 10.4 (SD) mmHg, respectively. The prevalence of hypertension was increased with age (p < 0.001). Of the 1109 (27.3%) hypertensive participants, 731 (65.9%) were aware of their hypertension. As shown in Fig 3, mean hip and waist circumferences were less in men than in women. The most prevalent self-reported

**Table 5. Baseline socio-demographic characteristics of participants.**

| Variable | No (%) | Mean ± SD |
|---|---|---|
| Sex | | |
| Male | 1725 (42.5) | |
| Female | 2334 (57.5) | |
| Age Group | | |
| 35–39 yr | 940 (23.2) | |
| 40–44 yr | 747 (18.4) | |
| 45–49 yr | 689 (17) | |
| 50–54 yr | 556 (13.7) | |
| 55–59 yr | 510 (12.6) | |
| 60–64 | 370 (9.1) | |
| >= 65 yr | 247 (6.1) | |
| Age | | 48.25 ± 9.4 |
| Education | | |
| illiterate | 1798 (44.3) | |
| Primary School | 954 (23.5) | |
| Secondary School | 511 (12.6) | |
| Diploma | 460 (11.3) | |
| University | 336 (8.28) | |
| Education Years | | 5.82 ± 4.8 |
| Marital Status | | |
| Single | 98 (2.4) | |
| Married | 3631 (89.5) | |
| Widowed | 254 (6.3) | |
| Divorced | 76 (1.9) | |
| Age at first marriage | | 20.85 ± 5.2 |
| Marriage with Relatives | | |
| No | 2626 (64.7) | |
| Yes—with first cousin | 734 (18.1) | |
| Yes—with second cousin | 598 (14.7) | |
| Missing | 101 (2.5) | |
| Employment | | |
| Employed | 1794 (44.2) | |
| Unemployed | 329 (8.1) | |
| Housewife | 1936 (47.7) | |
| Housing Ownership | | |
| Owned | 3439 (84.7) | |
| Leased | 333 (8.2) | |
| Other | 262 (6.5) | |
| Missing | 25 (0.6) | |
| Floor area of residence | | |
| less than 100 m$^2$ | 1386 (34.1) | |
| 100–200 m$^2$ | 2231 (55.0) | |
| more than 200 m$^2$ | 417 (10.3) | |
| Missing | 25 (0.6) | |
| Residential Status | | |
| urban | 3442 (84.8) | |
| rural | 617 (15.2) | |

**Table 6. Participants' baseline characteristics by gender and prevalence of self-reported health conditions.**

| Domain | Variable | Men | Women | P-value[1] |
|---|---|---|---|---|
| Lifestyle | Cigarette smoking | 33.8% | 0.3% | <0.001 |
| | Hookah smoking | 23.2% | 17.3% | <0.001 |
| | Alcohol drinking | 12.5% | 0.2% | <0.001 |
| | Drug use | 9.7% | 0.5% | <0.001 |
| | No toothbrushing | 6.5% | 3.7% | <0.001 |
| Anthropometrics | BMI (Mean ± SD) | 25.8 ± 4.5 | 27.8 ± 5.2 | <0.001 |
| | Underweight | 5.6% | 3.1% | <0.001 |
| | Normal | 43.0% | 29.5% | |
| | Overweight | 35.4% | 37.0% | |
| | Obese | 16.1% | 30.4% | |
| Physical Examination | Systolic Blood Pressure (Mean ± SD) | 120.51 ± 16.5 | 117.23 ± 18 | <0.001 |
| | Diastolic Blood Pressure (Mean ± SD) | 78.40 ± 9.9 | 75.77 ± 10.8 | <0.001 |
| Laboratory Findings | Fasting Blood Sugar (Mean ± SD) | 105.62 ± 37.7 | 109.96 ± 46.4 | <0.001 |
| | Total Cholesterol (Mean ± SD) | 198.99 ± 42.3 | 204.79 ± 42.3 | <0.001 |
| | TG (mg/dL) | 148.98 ± 42.3 | 127.23 ± 67.9 | <0.001 |
| | HDL (mg/dL) | 44.41 ± 9.6 | 50.48 ± 10.9 | <0.001 |
| | LDL (mg/dL) | 125.15 ± 33.3 | 129 ± 35.7 | <0.001 |
| Self-reported disease (diagnosed by physician) | Diabetes mellitus | 17.4% | 12.4% | <0.001 |
| | Hypertension | 14.6% | 21.0% | <0.001 |
| | MI | 2.6% | 1.0% | <0.001 |
| | Stroke | 0.8% | 0.9% | 0.725 |
| | Renal Failure | 0.4% | 0.8% | 0.109 |
| | Fatty Liver | 12.0% | 11.1% | 0.358 |
| | Thyroid | 3.2% | 13.9% | <0.001 |
| | Rheumatic disease | 1.1% | 5.5% | <0.001 |
| | Chronic lung disease (Asthma/ Tuberculosis) | 2.5% | 3.7% | 0.034 |
| | Breast cancer | - - | 0.6% | - - |
| | Cervical cancer | - - | 0.2% | - - |
| | Prostate cancer | 0.0% | - - | - - |
| | Colorectal cancer | 0.2% | 0.1% | 0.229 |
| | Depression | 0.8% | 1.7% | 0.008 |
| | Psychiatric disorders | 2.0% | 3.4% | 0.006 |
| | Osteoporosis | 20 (0.5%) | 262 (6.5%) | <0.001 |
| | Fracture | 106 (2.6%) | 141(3.5%) | 0.947 |

[1]The independent samples t-test was used to determine if there is a significant difference between the means of two groups, the chi-squared test was used to determine whether there is an association between 2 categorical variables.

NCDs were listed in Table 6. We found that the majority of reported NCDs were more prevalent in women than in men. Disparities in health due to differences in socioeconomic status (SES) was identified as a strong social determinant of health. In the present study, the prevalence of multimorbidity of 3 or more NCDs were 8% (No. 326) and 6% (No.240), respectively. Association of the co-occurrence of obesity risk factors with NCD multimorbidity was more significant (clustering of 3 and more NCDs as 6% and 5%) compared with co-existence of high cholesterol (clustering of 3 and more NCDs as 1.5% and 1%) and smoking (clustering of 3 and more NCDs as 1,25% and 0.8%) risk factors with multimorbidity. Consistent with the literature [28–30], we calculated SES index based on owning car, car price, house ownership, house

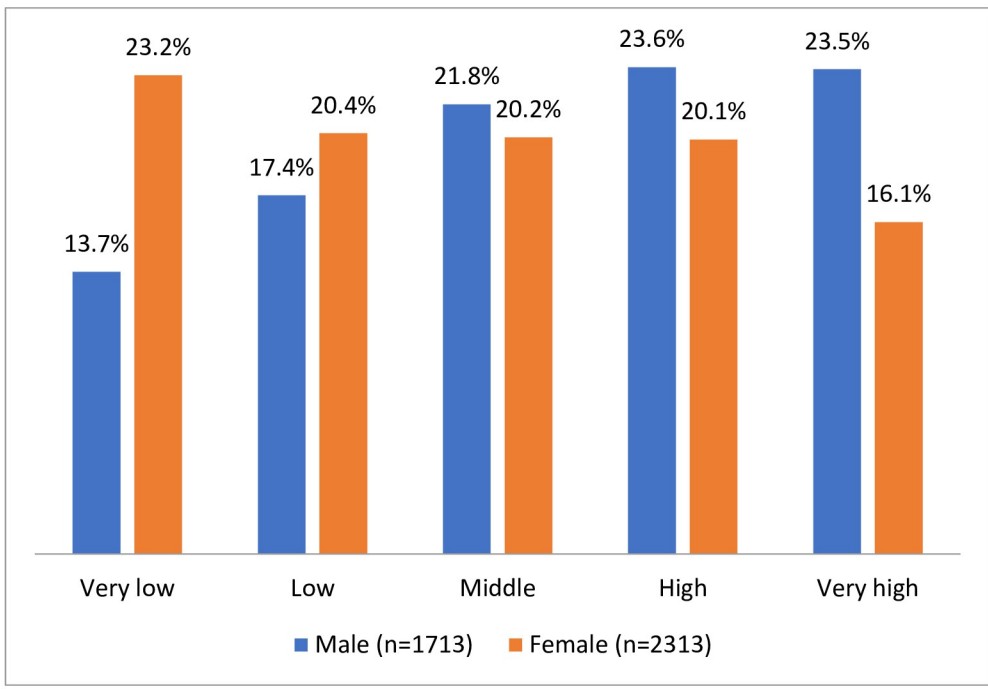

**Fig 3. Socioeconomic status of population in the cohort of Bandar Kong.**

area, owning computer (PC & Notebook), owning Dishwasher machine, number of travels per year and job title categorized into five groups, as shown in Fig 3. By Dec 30, 2019, we completed the third wave of follow-up with a participation above 99%. An overview of outcomes has been displayed in Table 7.

## Determinates of some of factors affecting hypertension, diabetes, and cardiac vascular disease

Association between the increase of risk of hypertension and age, being married, being a widow/divorced, and living in a rural area was found by adjusted ORs (Table 8). Based on Table 2, the aged 55–64 people were more likely to be at risk of hypertension compared to the aged 35–44 people (AOR = 1.30, 95% CI: 1.07,1.57). The odds to be at risk of hypertension was

**Table 7. The most common chronic NCDs diagnosed during the first and second annual follow-ups.**

| Chronic NCDs | 1$^{st}$ annual follow-up (%) | 2$^{nd}$ annual follow-ups (%) |
|---|---|---|
| Hypertension | 68 (37%) | 72 (53%) |
| Diabetes | 48 (26%) | 29 (21%) |
| Ischemic Heart disease | 17 (9.4%) | 11(8%) |
| Stroke | 15 (8.3%) | 10 (7%) |
| Cancers | 7 (3.8%) | 1 (0.7%) |
| COPD | 6 (3.3%) | 4 (3%) |
| Renal failure | 1 (0.5%) | - |
| Others | 6 (3.3%) | 1 (0.7%) |
| Death (Due to NCDS) | 12 (6.6%) | 14 (10%) |
| Total outcome | 180 | 136 |

**Table 8. Multivariable analysis of some of the factors affecting hypertension, diabetes, and heart diseases in the current study.**

| Variables | | | Hypertension[1] | | Diabetes[2] | | Cardiac Vascular Disease[3] | |
|---|---|---|---|---|---|---|---|---|
| | | | *Adjusted OR (95% CI) | P-value | *Adjusted OR (95% CI) | P-value | *Adjusted OR (95% CI) | P-value |
| Sex | (Ref: male) | | 0.94 (0.81,1.208) | 0.391 | 1.38(1.16,1.64) | <0.001[+] | 0.83(0.65,1.04) | 0.118 |
| Age | (Ref: 35–44) | 45–54 | 1.03(0.87,1.23) | 0.679 | 1.04(0.85,1.27) | 0.684 | 1.14(0.86,1.52) | 0.348 |
| | | 55–64 | 1.30(1.07,1.57) | 0.008[+] | 1.09(0.87,1.38) | 0.427 | 1.58(1.17,2.13) | 0.003[+] |
| | | ≥65 | 1.31(0.96,1.79) | 0.084 | 1.02(0.70,1.49) | 0.888 | 1.35(0.84,2.17) | 0.215 |
| Marital status | (Ref: single) | Married | 1.81(1.06,3.09) | 0.030[+] | 2.02(1.04,3.94) | 0.038[+] | 2.71(0.85,8.67) | 0.091 |
| | | Widow/divorced | 2.93(1.64,5.22) | <0.001[+] | 2.59(1.27,5.28) | 0.008[+] | 4.42(1.33,14.72) | 0.015[+] |
| Education | (Ref: >12years) | ≤5 years | 1.05(0.80,1.38) | 0.707 | 1.06(0.76,1.47) | 0.706 | 1.45(0.90,2.34) | 0.122 |
| | | 6-12years | 0.86(0.64,1.15) | 0.311 | 1.04(0.74,1.47) | 0.808 | 1.05(0.63,1.76) | 0.839 |
| Residence | (Ref: Urban) | | 1.33(1.09,1.61) | 0.004[+] | 1.28(1.02,1.61) | 0.027[+] | 0.91(0.66,1.25) | 0.576 |
| BMI | (Ref:18.5–24.9) | ≤ 18.4 | 0.71 (0.44,1.16) | 0.176 | 0.90(0.52,1.55) | 0.709 | 1.44(0.74,2.82) | 0.280 |
| | | 25.0–29.9 | 1.17(0.99,1.39) | 0.053 | 1.13(0.93,1.38) | 0.199 | 1.41(1.07,1.85) | 0.014[+] |
| | | 30.0–34.9 | 1.06(0.86,1.31) | 0.571 | 0.96(0.74,1.23) | 0.759 | 1.46(1.05,2.03) | 0.023[+] |
| | | ≥35 | 1.12(0.82,1.53) | 0.446 | 1.28(0.91,1.82) | 0.153 | 1.43(0.89,2.30) | 0.136 |
| Smoke Cigarette | (Ref:Non-smoker) | Current | 0.92(0.71,1.20) | 0.555 | 1.07(0.79,1.44) | 0.633 | 1.03(0.69,1.55) | 0.850 |
| | | ex-smoker | 1.11(0.82,1.50) | 0.467 | 1.21(0.86,1.71) | 0.258 | 0.72(0.42,1.23) | 0.240 |
| Physical activity (Daily MET)[++] | (Ref: ≥45) | 24–36.5 | 1.03(0.81,1.29) | 0.831 | 1.22(0.92,1.62) | 0.154 | 1.36(0.92,2.00) | 0.115 |
| | | 36.6–44.9 | 1.03(0.83,1.27) | 0.771 | 1.10(0.86,1.42) | 0.431 | 1.16(0.81,1.65) | 0.407 |

*Adjusted Odds Ratio.

[+] Significant at 0.05 level.

[++]Metabolic Equivalent of Task.

[1]Self-reported hypertension (HTN), anti-hypertensive medicine use, or a new hypertensive case were all used to define hypertension (HTN). People with a SBP of 140 mmHg or a DBP of 80 mmHg, or both, were considered as new hypertensive cases.

[2]T2DM was defined by the American Diabetes Association (ADA) as FPG 126 mg/dl in the first evaluation phase of study or the use of anti-hyperglycemic medications or self-reported disease.

[3]Cardiovascular disease (CVD) is defined by the presence coronary heart disease (CHD) or stroke by self-reporting. By definition, Osteoporosis and fracture were self-reported.

1.81 and 2.93 times higher for married and widowed/divorced people than the single ones, respectively (AOR = 1.81, 95% CI: 1.06,3.09; AOR = 2.93, 95% CI: 1.64,5.22). Besides, living in a rural area increased the odds of having hypertension by 1.33 (AOR = 1.33,95% CI: 1.09,1.61).

Moreover, association between an increase of risk of diabetes and being female, being married, being a widow/divorced, and living in a rural area was found. By adjusting the effect of other variables, being female was related to a higher risk of diabetes (AOR = 1.38, 95% CI: 1.16,1.64). The odds to be at risk of diabetes was 2.02 and 2.59 times higher for married and widowed/divorced people than the single ones, respectively (AOR = 2.02, 95% CI: 1.04,3.94; AOR = 2.93, 95% CI: 1.27,5.28). Moreover, living in a rural area increased the odds of having hypertension by 1.33 (AOR = 1.33,95% CI: 1.09,1.61).

In addition, age, being a widow/divorced and BMI increased the odds of having cardiac vascular disease. By controlling the effect of other variables, the aged 55–64 people were more likely to be at risk of cardiac vascular disease compared to the aged 35–44 people (AOR = 1.58, 95% CI: 1.17,2.13). Besides, the odds of having cardiac vascular disease was 4.42 times higher for widowed/divorced people than the single ones (AOR = 4.42, 95% CI: 1.33,14.72). Furthermore, the odds of having the cardiac vascular disease was higher for people who had BMI≥25

than people with 18.5≤BMI≤ 24.9. Lower Cardiovascular diseases, serum level of FBS and higher HDL level in sailors/fishermen compared to other job groups were significant (p-value <0.001) along with more findings in detailed have been represented and summarized in Table 9.

As observed in Table 10, intake of fruit, vegetables, and dairy was less than two servings per day in 9.2%, 13%, and 58.3% of the participants, respectively. Eating less than one serving of fish per week was reported by 6.3% of the participants. Soft drinks were served more than two servings per week by 79.7% and 53.6% had more than four servings of sugar per day. Intake of palm oil was equal or more than one serving per day in 26.9% of the participants; 75.3% consumed date more than two servings per day.

## Strengths and limitations

Here, in order to improve the external validity and to increase the generalizability of the study results to the target community, the participants were enrolled from the both gender groups with diverse education and jobs as well as various socio-economic status.

The most important strengths of the study include: a) The first population-based prospective cohort investigation of NCDs in Southern coastlines, b) Establish of valuable infrastructure for future large studies, c) Prospective investigation on lifestyle and risk factors to NCDs in sailors/fishermen and Islands inhabitants through well-trained staff under close supervision, d) set up a large biobank of various specimens to multi-omics study, e) minimum loss to follow-up f) connecting with corresponding electronic health records (SIB system), g) being open cohort, h) extensive data collection at baseline. Our valuable experience may serve as a model for neighborhood countries in the Middle East and North Africa harboring similar climate, socioeconomic, cultural and life style status.

We realized the weaknesses of the study as follows: a) Sample size, b) Budget limitation, c) Cultural and religious restrictions to collect data on sexual transmitted diseases, d) lower social activities of housebound individuals, e) little information on family tree f) failure to collect stool to study the impact of gut microbiota in NCDs development, g) limited to participants aged 35–70, h) information bias due to self-reported questionnaire.

## Conclusion

Kong cohort study is the first and unique ongoing prospective population-based cohort study which has been undertaken in sea ports and coastal areas of southern Iran. Iran has undergone a period of demographic, health and life style transition. Prospective studies are the ideal tools to follow these transitions and their impacts on health status. Kong cohort study with well-defined roadmap and standard protocols, suitable data quality and persistent annual follow-ups in southern Iran provided proper setting for investigating NCDs.

Noteworthy, the participation and response rates were 94% and 99%, respectively, as compared to participation rate of 70% in the Golestan, 94.9% in the Yazd and 57.5% in the Tehran Lipid and Glucose (TCS) cohort studies [13, 31, 32]. Current study revealed that compared to men, physical inactivity, higher BMI, overweight and obesity were more prevalent in women, the same as reported worldwide [33–35]. Unhealthy life style including physical inactivity and smoking was common in southern Iran. In the present study, the alteration of nutrition pattern of the participants was reflected in the low intake of fruit, vegetables, dairy and fish associated with high servings of palm oil and sugar. These findings are in concordance with high incidence of NCDs in Kong cohort study and southern Iran.

We recorded 28 deaths during first and second annual follow-ups till now. The main point is the higher prevalence of cardiovascular diseases including above 40% of the recorded deaths.

**Table 9. Comparison of major clinical and laboratory findings among jobless people, sailors/fishermen, employees and other occupations.**

| | Jobless[1] (n = 2254) | Sailor/Fisherman (n = 340) | Employee (n = 295) | Other jobs (n = 1157) | P- value[2] |
|---|---|---|---|---|---|
| Clinical findings | | | | | |
| Cardiac Disease, n (%) | 229(10.2) | 16(4.7) | 10(3.4) | 68(5.9) | <0.001 |
| MI, n (%) | 46(2.0) | 6(1.8) | 2(0.7) | 18(1.6) | 0.353 |
| BPS(mean ± SD) | 119.49±18.69 | 118.44±14.43 | 114.07±13.96 | 118.34±16.39 | <0.001 |
| BPD (mean ± SD) | 76.93±10.79 | 77.74±9.25 | 75.65±10.08 | 76.88±10.12 | 0.092 |
| Blood Analysis (mean ± SD) | | | | | |
| FBS | 111.50±47.65 | 101.26±31.09 | 100.42±26.52 | 105.56±38.84 | <0.001 |
| Cholesterol | 203.97±42.98 | 201.28±49.01 | 202.74±36.76 | 199.47±40.20 | 0.031 |
| Triglycerides | 130.36±70.76 | 156.03±158.24 | 140.74±94.55 | 141.86±85.22 | <0.001 |
| LDL cholesterol | 128.50±36.16 | 126.64±31.81 | 128.00±30.12 | 125.84±33.20 | 0.186 |
| HDL cholesterol | 49.63±10.89 | 44.49±9.46 | 46.65±10.14 | 46.01±10.46 | <0.001 |
| S.G.O.T (AST) | 20.89±11.08 | 23.76±9.45 | 24.29±15.53 | 22.87±10.10 | <0.001 |
| S.G.P.T (ALT) | 23.77±19.02 | 33.78±22.99 | 33.88±27.58 | 30.16±22.26 | <0.001 |
| Alkaline Phosphatase | 203.84±70.85 | 197.99±50.39 | 180.50±47.16 | 199.19±66.85 | <0.001 |
| Urine Analysis | | | | | |
| Blood (positive), n (%) | 360(16.1) | 34(10.1) | 23(7.9) | 136(11.8) | <0.001 |
| Protein (positive), n (%) | 125(5.6) | 16(4.8) | 5(1.7) | 46(5.0) | <0.001 |
| Glucose (positive),n(%) | 235(10.5) | 16(4.8) | 15(5.2) | 92(8.0) | <0.001 |

[1] Unemployed / Housewife / Retired;

[2] ANOVA was used to determine whether there are any statistically significant differences between the means of a continuous variable in 4 groups, and the chi-squared test was used to determine whether there is an association between 2 categorical variables.

These findings are consistent with national estimates. Here, Women's participation rate was higher, the same as happened in PERSIAN Cohort Studies, however, at baseline, we observed diverse demographic and clinical characteristics in current study as compared with other national cohorts. For instance, the prevalence of alcohol drinking was higher in Kong cohort compared to Golestan and TCS (12.5%, 0.7% and 7.9%, respectively) [28, 29]. Of note, our study showed higher prevalence of diabetes mellitus, while lower of that for hypertension as compared to Pars cohort study (17.4%, 9.4% for diabetes and 26.9%, 21% for hypertension, respectively) [36]. The prevalence of NCD multimorbidity is growing in low- and middle-income countries and being estimated 7.8%. In future, multimorbidity will be an extreme challenge for both developed and developing countries [37, 38]. Awareness on multimorbidity patterns is of important implications for prevention, diagnosis, and management. In the present

**Table 10. Baseline dietary intake of participants in Bandar Kong cohort study.**

| | All | Male, N (%) | Female, N (%) | P value |
|---|---|---|---|---|
| Eating less than two servings of fruits per day | 371 (9.2) | 121 (7.1) | 250 (10.8) | <0.001 |
| Eating less than two servings of vegetables per day | 524 (13) | 196 (11.4) | 328 (14.2) | 0.011 |
| Eating less than two servings of dairy per day | 2353 (58.3) | 870 (50.7) | 1483 (64) | <0.001 |
| Eating less than one serving of fish per week | 255 (6.3) | 74 (4.3) | 181 (7.8) | <0.001 |
| Eating more than two servings of soft drink per week | 3212 (79.7) | 1162 (67.7) | 2050 (88.5) | <0.001 |
| Eating more than four servings of sugar per day | 2162 (53.6) | 1036 (60.4) | 1126 (48.6) | <0.001 |
| Eating equal or more than one serving of palm oil per day | 1083 (26.9) | 462 (26.9) | 621 (26.8) | 0.938 |
| Eating more than two servings of date per day | 3038 (75.3) | 1350 (78.8) | 1688 (78.9) | <0.001 |

study, our epidemiologic data revealed a high NCD multimorbidity that could be implicated in health policies and developing guidelines and designing strategies to reduce the burden of the disease.

The ongoing longitudinal cohort possesses the potential to survey and evaluate the lifestyle changes over the lifetime. We planned to increase the sample size and improve the cohort quality during annual follow-ups to generate fresh and up to date data for tackling NCDs. The key justification for running this cohort was to cover various geographical regions including mountains and coastlines in vicinity of each other with hot and humid climate almost throughout the year, which made it a unique environment likely leading to development of region-specific risk factors and NCDs.

In this study, the primary findings demonstrated high prevalence of behavioral modifiable risk factors. Therefore, conducting population-specific interventions plus nested case-control studies along with persistent monitoring the trend of development of chronic diseases could generate valuable evidence-based policies in controlling the risk factors. Prevention of premature deaths due to NCD and decrease of associated health care expenses will be the main objectives of health policy. The consequences of such investigations would aid policy makers to plan and implement cost-effective facility within the situation of primary health care in Iran, which could serve as a model for neighboring countries through.

At last, we effectively established a population-based prospective cohort of adults that highlighted the requirement for evolving NCD intervention programs, and launching active surveillance for operative planning, application and evaluation. Comprehensive control of NCD risk factors has turn into an absolute requisite worldwide. Blood specimens collected at baseline and in each follow-up will be a distinctive and potent resource for investigating the association between major risk factors with NCDs and exploring further original biomarkers. The ultimate goal is sustaining a healthy life style which would promise prevention and reduction the burden of NCDs.

## Supporting information

**S1 File. Questionnaire.**
(PDF)

**S2 File. Data dictionary for baseline variables.**
(PDF)

**S3 File. Rationale, objectives, and design.**
(PDF)

**S4 File. Questionnaire in original language.**
(PDF)

## Acknowledgments

First, we would like to express our sincere gratitude to all participants took part in the study. People of Bandar Kong helped us a lot to initiate and maintain the study; thanks a lot, to all of them. In fact, without help and assistance of cohort staff, health workers, regional authorities, mayor of the district, the work did not pursue, our appreciations to all. The last but not the least, professor Reza Malekzadeh, Director of Digestive Diseases Research Institute, Tehran University of Medical Sciences, who really encouraged and supported me as principal investigator to launch the study in southern Iran, special gratefulness to him.

## Author Contributions

**Conceptualization:** Hossein Poustchi.

**Data curation:** Azim Nejatizadeh, Ebrahim Eftekhar, Amin Ghanbarnejad, Shideh Rafati.

**Formal analysis:** Amin Ghanbarnejad, Shideh Rafati.

**Funding acquisition:** Azim Nejatizadeh, Mohammad Shekari, Hossein Farshidi, Hossein Poustchi, Teymour Aghamolaei.

**Investigation:** Azim Nejatizadeh, Mehdi Shahmoradi, Hossein Poustchi, Hadi Yousefi.

**Methodology:** Azim Nejatizadeh, Ebrahim Eftekhar, Mehdi Shahmoradi, Hossein Poustchi, Amin Ghanbarnejad, Hadi Yousefi, Shideh Rafati.

**Project administration:** Azim Nejatizadeh, Mohammad Shekari, Hossein Farshidi, Seyed Hossein Davoodi, Mehdi Shahmoradi, Hossein Poustchi, Teymour Aghamolaei.

**Resources:** Azim Nejatizadeh, Mohammad Shekari, Seyed Hossein Davoodi, Mehdi Shahmoradi, Hossein Poustchi, Teymour Aghamolaei.

**Software:** Hadi Yousefi, Shideh Rafati.

**Supervision:** Azim Nejatizadeh, Ebrahim Eftekhar, Hossein Farshidi, Seyed Hossein Davoodi, Mehdi Shahmoradi, Hossein Poustchi, Teymour Aghamolaei, Hadi Yousefi.

**Validation:** Ebrahim Eftekhar, Amin Ghanbarnejad, Hadi Yousefi, Shideh Rafati.

**Visualization:** Azim Nejatizadeh, Ebrahim Eftekhar, Mohammad Shekari, Hossein Farshidi, Seyed Hossein Davoodi, Mehdi Shahmoradi.

**Writing – original draft:** Azim Nejatizadeh.

**Writing – review & editing:** Azim Nejatizadeh, Seyed Hossein Davoodi.

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
