## [Decision Letter · Decision Letter 0]

10 Nov 2020

PONE-D-20-32677

Cohort profile: The Bandar Kong prospective cohort study

PLOS ONE

Dear Dr. Azim

Thank you for submitting your manuscript to PLOS ONE. After careful consideration, we feel that it has merit but does not fully meet PLOS ONE’s publication criteria as it currently stands. Therefore, we invite you to submit a revised version of the manuscript that addresses the points raised during the review process.

There are many concerns including the available other cohorts and reports that you should discuss in your report for comparison purpose. Other reviewers also mentioned about the ration of men to female in Iran that you need to justify.

We look forward to receiving your revised manuscript.

Kind regards,

Hassan Ashktorab

Academic Editor

PLOS ONE

Journal Requirements:

3. Please include a copy of the questionnaire in the original language, as Supporting Information, or include a citation if it has been published previously.

4. In statistical methods, please clarify whether you corrected for multiple comparisons.

5. In your statistical analyses, please state whether you accounted for repeated measurements in your statistical models.

6. We note that you have indicated that data from this study are available upon request. PLOS only allows data to be available upon request if there are legal or ethical restrictions on sharing data publicly. For information on unacceptable data access restrictions, please see http://journals.plos.org/plosone/s/data-availability#loc-unacceptable-data-access-restrictions.

7. We note that Figure 1 in your submission contains map images which may be copyrighted.

We require you to either (a) present written permission from the copyright holder to publish this figure specifically under the CC BY 4.0 license, or (b) remove the figure from your submission:

b. If you are unable to obtain permission from the original copyright holder to publish this figure under the CC BY 4.0 license or if the copyright holder’s requirements are incompatible with the CC BY 4.0 license, please either i) remove the figure or ii) supply a replacement figure that complies with the CC BY 4.0 license. Please check copyright information on all replacement figures and update the figure caption with source information. If applicable, please specify in the figure caption text when a figure is similar but not identical to the original image and is therefore for illustrative purposes only.

8. PLOS requires an ORCID iD for the corresponding author in Editorial Manager on papers submitted after December 6th, 2016. Please ensure that you have an ORCID iD and that it is validated in Editorial Manager. To do this, go to ‘Update my Information’ (in the upper left-hand corner of the main menu), and click on the Fetch/Validate link next to the ORCID field. This will take you to the ORCID site and allow you to create a new iD or authenticate a pre-existing iD in Editorial Manager. Please see the following video for instructions on linking an ORCID iD to your Editorial Manager account: https://www.youtube.com/watch?v=_xcclfuvtxQ

Reviewers' comments:

Reviewer's Responses to Questions

**Comments to the Author**

1. Is the manuscript technically sound, and do the data support the conclusions?

Reviewer #1: Partly

Reviewer #2: Yes

2. Has the statistical analysis been performed appropriately and rigorously? 

Reviewer #1: No

Reviewer #2: I Don't Know

3. Have the authors made all data underlying the findings in their manuscript fully available?

Reviewer #1: Yes

Reviewer #2: Yes

4. Is the manuscript presented in an intelligible fashion and written in standard English?

Reviewer #1: No

Reviewer #2: Yes

5. Review Comments to the Author

Reviewer #1: Dear respected authors,

The authors of this manuscript present the results of about two years precise work to launch a prospective cohort study in southern Iran. The efforts of these colleagues and their supporter team in the Ministry of Health are commendable.

Please note the following:

1- I suggest that the manuscript should be read by a native English language editor to check and correct the grammatical mistakes. For example, in line 27, “in” was repeated twice, or at the beginning of many sentences, digits were seen, which must be written in letters, and ...

2- According to the 2016 census in I R of Iran, the M/F ratio in Hormozgan province was 51% to 49%, in comparison the ratio in this cohort study, where the ratio of men to women is 42.5% to 57.5%. There is a big difference. The study population may not represent the main population in the province.

3- In general, the manuscript is too long. It is better to reduce words to about 3000 to 3500 .

4- Comparison of significance level between men and women in cohort profile manuscripts is usually not required.

5- It is written in line 152: At the beginning, and scientific committee…. Which does not seem to be correct.

6- What were the selection criteria of 4200 participants out of about 6000 people at the beginning of the study?

7. Are the demographic differences between those who participated in the study and those who did not, compared?

8. Line 234, what diagnostic code is used in the study.

9- In the statistical analysis section, several tests are mentioned, but there is no report of some of them in the result section. Of course, there is no need for statistical analysis in the cohort profile manuscripts.

Reviewer #2: this research is unique in south of Iran but In case of statistical analysis, an expert scientist should evaluate the results, however, there are several cohort study in Iran. I think it is worth nothing that this manuscript describe this types of diseases in specific part of Iran and compared with other region of Iran.

6. PLOS authors have the option to publish the peer review history of their article (what does this mean?). If published, this will include your full peer review and any attached files.

Reviewer #1: No

Reviewer #2: No

---

## [Author Response · Author response to Decision Letter 0]

1 Apr 2021

The Editor-in-chief 

PLOS ONE

With respect,

Responses to the referee's comments:

Manuscript Number: PONE-D-20-32677

Title: Reply to Manuscript Number: PONE-D-20-32677Title: Cohort Profile: The Bandar Kong Prospective Cohort Study

Article Type: Original research paper (Cohort profile)

Thanks a lot for your critical and peer review of our manuscript (ms). Our manuscript has been accepted with revision. We have provided a point-by-point response to all comments from the editors and reviewers and have revised the manuscript accordingly. 

Therefore, we resubmit our manuscript (ms) titled “Cohort Profile: The Bandar Kong Prospective Cohort Study”, authored by Azim Nejatizadeh et al. Efforts have been made to keep the text to minimum with inclusion of the most relevant references. We edited the text and made the corrections and needful modifications as per the reviewer comments. In fact, we shortened the manuscript drastically and made well-defined focus objectives. 

We have incorporated the editorial corrections in the revised version. 

Responses to the Journal Requirements:

Requirement 1: Please ensure that your manuscript meets PLOS ONE's style requirements, including those for file naming.

Reply 1: We appreciate your suggestion. Of course, we followed the journal’s style requirements, such as file naming and so on. 

Requirement 2: We suggest you thoroughly copyedit your manuscript for language usage, spelling, and grammar.

Reply 2: Thanks for the valuable comment. We do agree with the point which you raised. We took your point into consideration and modified/copyedited the entire text meticulously, spelling and then corrected grammar errors. We kept the manuscript as short as possible. We are not averse to further modifications of the language or the text. 

Requirement 3: Please include a copy of the questionnaire in the original language, as Supporting Information, or include a citation if it has been published previously.

Reply 3: We appreciate your suggestion. Earlier, we uploaded the English version of it as “S1 File. Questionnaire. (PDF)”. We take your point into consideration while submission of the manuscript and will add and upload a copy of original language of the questionnaire in Persian. 

Requirement 4: In statistical methods, please clarify whether you corrected for multiple comparisons.

Reply 4: I would like to take your kind consideration to the point that multiple correction is used mainly in a group of conditions to correct experiment-wise error rate once applying multiple “t” tests or as a post-hoc procedure to correct the cluster-wise error rate subsequent to analysis of variance (ANOVA). Rather, that is an adjustment to restrict the state of false positive and to avoid data from mistakenly looking to be statistically significant. Therefore, we didn’t correct for multiple comparison, since this is a cohort profile and we only reported the main characteristics of the population.

Requirement 5: In your statistical analyses, please state whether you accounted for repeated measurements in your statistical models.

Reply 5: Thanks a lot for raising this issue. To the best of my knowledge, the prospective cohort studies mainly underpin investigating the relation between exposure measure at baseline and occurrence of the health outcome of interest throughout follow up period. Repeated measurements can be made of both exposures and outcomes. However, at the time being, we completed the enrollment phase and started follow up phases. Of note, the submitted manuscript is just the cohort profile of the ongoing study and now, we used the data on baseline for the cohort profile and therefore we didn’t perform repeated measurement analysis.

Requirement 6: We note that you have indicated that data from this study are available upon request. PLOS only allows data to be available upon request if there are legal or ethical restrictions on sharing data publicly. 

Reply 6: Yes. As you have noted, we have incorporated a section referred to as “Access to the data” at the end of the manuscript which has been set according to the Journal instructions to the authors. The respective section has also appeared below for your kind consideration. 

 Access to the data 

The study is a part of the PERSIAN Cohort Study, the kind applicants should follow the respective rules and the protocol of data sharing and scientific collaboration. In case of any application or query, you may kindly email the principle investigator. All interested investigators in Iran and worldwide would have free access to the data of this study, and necessary processes are available at the Cohort website to reproduce the project, participate in collaborative research projects, and use the data. Access to the data is available from corresponding cohort (azimnejate@yahoo.com) or from http://persiancohort.com/access

Requirement 7: In your revised cover letter, please address the following prompts:

b) If there are no restrictions, please upload the minimal anonymized data set necessary to replicate your study findings as either Supporting Information files or to a stable, public repository and provide us with the relevant URLs, DOIs, or accession numbers. 

Reply 7: As we addressed the matter in “Reply to Requirement 6”, there are some legal restrictions on sharing a de-identified data set, if requested without prior permission. We emphasized that access to the data is available through corresponding author (Email: azimnejate@yahoo.com; Phone number of the cohort center: +98-7644235008) or via http://persiancohort.com/access , in order to get your request fulfilled. Furthermore, you communicate or contact the vice chancellery of Hormozgan university of medical sciences from http://resv.hums.ac.ir/ for further administrative processing of your application. 

Requirement 8: We note that Figure 1 in your submission contains map images which may be copyrighted.

Reply 8: In fact, the map, i.e. Figure 1, is a part of a detailed plan of Bandar Kong Municipality (https://www.cityhallworldwide.com/iran-town-hall/bandar-kong-municipality) designed and drawn by Hormozgan Road and Urban Development Administration, then approved by the Supreme Council of Architecture and Urban Planning of Iran (https://www.mrud.ir/). We described in “method and materials” section of the manuscript that the function and field activity of the cohort center was launched and coordinated through Bandar Kong Municipality as a main stakeholder and sponsor of the study. Of course, once the study started, the concerned authority provided us all the needful requirements including the maps and so on.

Regarding “Figure” representing the sites of PERSIAN Cohort as well as Kong Cohort Study across country (http://persiancohort.com/access). In introduction of the manuscript, we explained that our study is a part of PERSIAN Cohort study, that is why, we incorporated this map from PERSIAN Cohort. If you are not satisfied with the justification, you may remove this figure.

Requirement 9: PLOS requires an ORCID iD for the corresponding author in Editorial Manager on papers submitted after December 6th, 2016. Please ensure that you have an ORCID iD and that it is validated in Editorial Manager.

Reply 9: We appreciate your suggestion to ensure to have a validated ORCID iD. Of course, my ORCID iD is https://orcid.org/0000-0001-6347-0163. I checked it out and that is validated in Editorial Manager.

Reviewers'comments:

Reviewer#1: Dear respected authors,

The authors of this manuscript present the results of about two years precise work to launch a prospective cohort study in southern Iran. The efforts of these colleagues and their supporter team in the Ministry of Health are commendable.

Please note the following:

Comment 1: I suggest that the manuscript should be read by a native English language editor to check and correct the grammatical mistakes. For example, in line 27, “in” was repeated twice, or at the beginning of many sentences, digits were seen, which must be written in letters, and ...

Reply 1: We welcome your suggestion. With respect to your valuable comment, we corrected the grammatical errors throughout the text. As much as possible, we did our best to shorten the text given that the concept and the essence of the study under the defined objectives remain intact. Based on the principles of good practice for academic writing, all the authors with aid of English native colleagues who are scientist and expert in the field and academic writing, read the manuscript thoroughly and meticulously for several times. 

Comment 2: According to the 2016 census in I R of Iran, the M/F ratio in Hormozgan province was 51% to 49%, in comparison the ratio in this cohort study, where the ratio of men to women is 42.5% to 57.5%. There is a big difference. The study population may not represent the main population in the province.

Reply 2: We appreciate your meticulous comment. As we stated in the subheading “Strengths and Limitations”, the sample size is not that much large for a population-based prospective cohort; that is why, perhaps, this weakness being the argument to the observed difference. To tackle the issue, we started increasing the sample size twofold to normalize the male to female ratio as the main population of the province. Second, we realize that the province possesses a different cultural behavior and the major of the men are in sea-related industry such as fishing, ship making and etc. whereas most of the women are housewives. Therefore, the ratio of women participated in the study isn’t the same as the sex ratio of the province. Most probably, One of the main reasons is men’s employment and their limited time for participation.

Comment 3: In general, the manuscript is too long. It is better to reduce words to about 3000 to 3500.

Reply 3: We agree with your comment. We also like to keep the manuscript as short as possible, however, in this particular study, we have to maintain the concept and essence of the study according to the pre-defined objectives. Explaining the importance of each section was expected to give a look of lengthy text especially in the Method and materials. However, we revised/modified and tried our best to shorten it much more. Kindly note we have now reshaped and shortened the manuscript largely by deleting text where most appropriate as per the reviewers’ valuable suggestions. We really hope that you will be considerate to it and our explanations satisfy you. 

Comment 4: Comparison of significance level between men and women in cohort profile manuscripts is usually not required.

Reply 4: This has been done like some other studies listed below (1,2). However, if you think it should be removed to improve the manuscript, we will delete it.

1. Pasdar Y, Najafi F, Moradinazar M, Shakiba E, Karim H, Hamzeh B, Nelson M, Dobson A. Cohort profile: Ravansar Non-Communicable Disease cohort study: the first cohort study in a Kurdish population. International journal of epidemiology. 2019 Jun 1;48(3):682-3f.

2. Mirzaei M, Salehi-Abargouei A, Mirzaei M, Mohsenpour MA. Cohort Profile: The Yazd Health Study (YaHS): a population-based study of adults aged 20–70 years (study design and baseline population data). International journal of epidemiology. 2018 Jun 1;47(3):697-8h.

Comment 5: It is written in line 152: At the beginning, and scientific committee…. Which does not seem to be correct.

Reply 5: Thanks for your comment. We corrected that. 

Comment 6: What were the selection criteria of 4200 participants out of about 6000 people at the beginning of the study?

Reply 6: We have clarified that in the text as follows:

“At first, the scientific committee enrolled 4200 participants out of 6000 permanent residents aged 35-70 years subjects in accordance with PERSIAN Cohort agreement. The inclusion criteria were: (i) willingness to participation, (ii) aged 35-75, (iii) residency for one year and at least nine months each year, (iv) written informed consent, (v) Iranian national. Exclusion criteria included: (i) no interest to attend, (ii) being guest, and (iii) mental or physical disabilities.”

Comment 7: Are the demographic differences between those who participated in the study and those who did not, compared?

Reply 7: It has not done. We have no meticulous and detailed information about the people who didn’t participate in the study.

Comment 8: Line 234, what diagnostic code is used in the study.

Reply 8: As we stated in text, two external internists independently review all available clinical documents and allocate a disease code and a date of occurrence to each outcome. The two disease codes are compared, and if they are different, a third, more senior internist reviews the data and makes the final decision on the code. The “code” was assigned based on ICD10 (international classification of diseases).

Comment 9: In the statistical analysis section, several tests are mentioned, but there is no report of some of them in the result section. Of course, there is no need for statistical analysis in the cohort profile manuscripts.

Reply 9: We appreciate your comment and agree with that. Hence, we removed the unnecessary tests in this phase of the cohort study. Therefore, we modified statistical analysis as per your suggestion.

Reviewer #2: this research is unique in south of Iran but in case of statistical analysis, an expert scientist should evaluate the results, however, there are several cohort studies in Iran. I think it is worth nothing that this manuscript describes these types of diseases in specific part of Iran and compared with another region of Iran.

Reply: Thanks for your comments. 

First, we described in the text under the subheading of “Method and materials” that a panel of experts and academics (scientists) were engaged in the design and conducting the study. Furthermore, we included one epidemiologist and one biostatistician in the scientific committee so as they used to evaluate the results and monitor the progress of the ongoing cohort in a regular basis.

Second, as per your suggestion, we compared some of the important demographic and clinical characteristics at base line among current running various cohorts at national level, in the Discussion section.

Upon acceptance of the manuscript, the raw data will be made available at this site and the access would be free for all academic users under a click wrap protection.

We thank the reviewers and editorial team for helping us improve our manuscript.

We are highly hopeful that the modifications in the manuscript and our explanations satisfy the referees. Kindly note we have now reshaped the manuscript by adding and deleting text where most appropriate as per the reviewers’ valuable suggestions. 

However, we are afraid now with so many of your suggestions that have been incorporated in the text, it has shortened largely, although we revised and tried our best to shorten it much more. We really hope that you will be considerate to it.

We are not averse to further modifications of the language or the text. 

We are highly hopeful that the ms finds potential, could fulfill the high standards of the journal and gets positive review.

Warm regards,

Azim Nejatizadeh, MD., PhD.

Corresponding author

Hormozgan University of Medical Sciences,

Bandar Abbas, Iran

Mobile: +98-9179564291

E-mail: azimnejate@yahoo.com

---

## [Decision Letter · Decision Letter 1]

12 May 2021

PONE-D-20-32677R1

Cohort profile: The Bandar Kong prospective cohort study

PLOS ONE

Dear Dr. Nejatizadeh

Thank you for submitting your manuscript to PLOS ONE. After careful consideration, we feel that it has merit but does not fully meet PLOS ONE’s publication criteria as it currently stands. Therefore, we invite you to submit a revised version of the manuscript that addresses the points raised during the review process.

Hi 

One of the reviewer still has major problem with the statistic analysis.

Please revisit his concern as follow:

 For example, without using OR, it is mentioned again and sample size selection eligibility criteria. And the repeated-measurement phase. Also addressing the issue of external validity. How many strategies was used to control of many kind of bias in the study.

We look forward to receiving your revised manuscript.

Kind regards,

Hassan Ashktorab

Academic Editor

PLOS ONE

Reviewers' comments:

Reviewer's Responses to Questions

**Comments to the Author**

1. If the authors have adequately addressed your comments raised in a previous round of review and you feel that this manuscript is now acceptable for publication, you may indicate that here to bypass the “Comments to the Author” section, enter your conflict of interest statement in the “Confidential to Editor” section, and submit your "Accept" recommendation.

Reviewer #1: (No Response)

Reviewer #2: All comments have been addressed

2. Is the manuscript technically sound, and do the data support the conclusions?

Reviewer #1: Partly

Reviewer #2: Yes

3. Has the statistical analysis been performed appropriately and rigorously? 

Reviewer #1: N/A

Reviewer #2: Yes

4. Have the authors made all data underlying the findings in their manuscript fully available?

Reviewer #1: Yes

Reviewer #2: Yes

5. Is the manuscript presented in an intelligible fashion and written in standard English?

Reviewer #1: Yes

Reviewer #2: Yes

6. Review Comments to the Author

Reviewer #1: Dear Authors,

Thank you for paying attention to my comments. Some of my comments have not been corrected. Including in the statistical analysis section. For example, without using OR, it is mentioned again. In general, in my opinion, it is better to first try to publish this study as a protocol of the cohort, and then after obtaining more results and publishing articles from the study results, proceed to publish the profile. For instance, after the repeated-measurement phase.

Strengths and weaknesses: I would recommend addressing the issue of external validity. How many strategies was used to control of many kind of bias in the study.

Reviewer #2: Dear Authors,

I went through the all responses and revised manuscript, I think this format is proper for publication

7. PLOS authors have the option to publish the peer review history of their article (what does this mean?). If published, this will include your full peer review and any attached files.

Reviewer #1: No

Reviewer #2: No

---

## [Author Response · Author response to Decision Letter 1]

25 Jun 2021

The Editor-in-chief 

PLOS ONE

With respect,

Responses to the referee's comments:

Manuscript Number: PONE-D-20-32677R1

Title: Reply to Manuscript Number: PONE-D-20-32677R1

Title: Cohort Profile: The Bandar Kong Prospective Cohort Study

Article Type: Original research paper (Cohort profile)

Thanks a lot for your critical and peer review of our manuscript (ms). Our manuscript has been accepted with revision. We have provided a point-by-point response to all comments from the reviewer and have revised the manuscript accordingly. Therefore, we resubmit our manuscript (ms) titled “Cohort Profile: The Bandar Kong Prospective Cohort Study”, authored by Azim Nejatizadeh, et al. Efforts have been made to keep the text to minimum with inclusion of the most relevant references. We edited the text and made the corrections and needful modifications as per the reviewer's comments. Furthermore, we have incorporated the editorial corrections in the revised version. 

Academic Editor

One of the reviewers still has major problem with the statistical analysis; Please revisit his concern as follow:

For example, without using OR, it is mentioned again and sample size selection eligibility criteria. And the repeated-measurement phase. Also addressing the issue of external validity. How many strategies was used to control of many kind of bias in the study.

Reply: I would like to thank you again for raising and reflecting the reviewer's concern. 

1. At first, regarding OR, I explained and added more findings by performing multivariable analysis represented as "Adjusted OR (95%CI) and P-value" shown in Table 8 on some of factors affecting hypertension, diabetes, and heart diseases in the present study. We embedded the Table 8 within the manuscript followed by interpretation of the generated results as follow: 

Determinates of some of factors affecting hypertension, diabetes, and cardiac vascular disease: Association between the increase of risk of hypertension and age, being married, being a widow/divorced, and living in a rural area was found by adjusted ORs (Table 8). Based on table2, the aged 55-64 people were more likely to be at risk of hypertension compared to the aged 35-44 people (AOR=1.30, 95% CI: 1.07,1.57). The odds to be at risk of hypertension was 1.81 and 2.93 times higher for married and widowed/divorced people than the single ones, respectively (AOR =1.81, 95% CI: 1.06,3.09; AOR =2.93, 95% CI: 1.64,5.22). Also, living in a rural area increased the odds of having hypertension by 1.33 (AOR=1.33,95% CI: 1.09,1.61). 

Moreover, association between the increase of risk of diabetes and being female, being married, being a widow/divorced, and living in a rural area was found. By adjusting the effect of other variables, being female was related to a higher risk of diabetes (AOR=1.38, 95% CI: 1.16,1.64). The odds to be at risk of diabetes was 2.02 and 2.59 times higher for married and widowed/divorced people than the single ones, respectively (AOR =2.02, 95% CI: 1.04,3.94; AOR =2.93, 95% CI: 1.27,5.28). Also, living in a rural area increased the odds of having hypertension by 1.33 (AOR=1.33,95% CI: 1.09,1.61). 

 In addition, age, being a widow/divorced and BMI increased the odds of having cardiac vascular disease. By controlling the effect of other variables, the aged 55-64 people were more likely to be at risk of cardiac vascular disease compared to the aged 35-44 people (AOR=1.58, 95% CI: 1.17,2.13). Also, the odds of having cardiac vascular disease was 4.42 times higher for widowed/divorced people than the single ones (AOR=4.42, 95% CI: 1.33,14.72). Furthermore, the odds of having the cardiac vascular disease was higher for people who had BMI≥25 than people who had 18.5≤BMI≤ 24.9. 

2. Regarding "sample size selection eligibility", within the text under the subtitle " Study population" we explained that the scientific committee enrolled 4200 participants in accordance with PERSIAN Cohort agreement. Should the prevalence of the outcome in the exposed group is 20%, 270 cases of interest would offer 90% power to detect a relative risk of 2. To obtain 270 cases over a period of 5 years, only 54 cases of interest per year is needed. The inclusion criteria were: (i) willingness to participation, (ii) aged 35-75, (iii) residency for one year and at least nine months each year, (iv) written informed consent, (v) Iranian national. Exclusion criteria included: (i) no interest to attend, (ii) being guest, and (iii) mental or physical disabilities. Furthermore, we started increasing the sample size twofold to normalize the male to female ratio as the main population of the province.

3. "Regarding repeated measurements" it is worth clarifying that analysis can be used to assess changes over time in an outcome measured serially or to test for differences in 1 or more treatments based on repeated assessments in the same subjects. Hence, we planned and initiated to follow up the participants each year by phone to look for: a) to renew the phone calls and addresses, b) reexamination of past medical history, c) investigating any recent treatment, hospitalization, NCDs or death. Therefore, the whole recruitment phases and repeated measurements every 5-year for 15 years with annual follow-up according to the standard PERSIAN Cohort protocol. We, now, have completed the third annual follow-up and just started the second 5-year recruitment phase and repeated measurements. 

4. …. Also addressing the issue of external validity. How many strategies was used to control of many kind of bias in the study.

In this study, in order to improve the external validity and to increase the generalizability of the study results to the target community, we selected eligible individuals from both sex groups with diverse education and jobs as well as unlike socio-economic status to represent that the designated individuals to be the appropriate representative from the reference population. To control and monitor carefully the quality and accuracy of measuring devices such as scales and blood pressure, we calibrated all the concerned devices periodically according to the standard protocol. Furthermore, training and certifying the observers' skills and standardizing the measurement methods were the two other strategies to control information bias that we used to regularly carry out and practice these approaches in our ongoing cohort study. 

To the best of my knowledge, one practical method to identify confounders is to analyze data with or without potential confounders. If the estimation of the association among the independent variables and the outcomes varies considerably by controlling a third variable, then the third variable would be confounding. Therefore, in the present study, using statistical methods such as regression models, the relationship between the independent variable and the desired outcome in different categories of confounding variables is calculated separately or controlled by statistical methods such as analysis stratification. We really hope that you will be considerate to it and our explanations satisfy the Reviewer. 

Review Comments to the Author

Reviewer#1: Dear Authors,

Thank you for paying attention to my comments. Some of my comments have not been corrected. Including in the statistical analysis section. For example, without using OR, it is mentioned again. In general, in my opinion, it is better to first try to publish this study as a protocol of the cohort, and then after obtaining more results and publishing articles from the study results, proceed to publish the profile. For instance, after the repeated-measurement phase.

Strengths and weaknesses: I would recommend addressing the issue of external validity. How many strategies was used to control of many kinds of bias in the study.

Reply to Reviewer#1 comments: 

At first, regarding "OR", we explained and added more findings by performing multivariable analysis represented as "Adjusted OR (95%CI) and P-value" shown in Table 8 on some of factors affecting hypertension, diabetes, and heart diseases in the present study. We embedded Table 8 within the manuscript followed by interpretation of the generated results as follow: 

Determinates of some of factors affecting hypertension, diabetes, and cardiac vascular disease: Association between the increase of risk of hypertension and age, being married, being a widow/divorced, and living in a rural area was found by adjusted ORs (Table 8). Based on table2, the aged 55-64 people were more likely to be at risk of hypertension compared to the aged 35-44 people (AOR=1.30, 95% CI: 1.07,1.57). The odds to be at risk of hypertension was 1.81 and 2.93 times higher for married and widowed/divorced people than the single ones, respectively (AOR =1.81, 95% CI: 1.06,3.09; AOR =2.93, 95% CI: 1.64,5.22). Also, living in a rural area increased the odds of having hypertension by 1.33 (AOR=1.33,95% CI: 1.09,1.61). 

Moreover, association between the increase of risk of diabetes and being female, being married, being a widow/divorced, and living in a rural area was found. By adjusting the effect of other variables, being female was related to a higher risk of diabetes (AOR=1.38, 95% CI: 1.16,1.64). The odds to be at risk of diabetes was 2.02 and 2.59 times higher for married and widowed/divorced people than the single ones, respectively (AOR =2.02, 95% CI: 1.04,3.94; AOR =2.93, 95% CI: 1.27,5.28). Also, living in a rural area increased the odds of having hypertension by 1.33 (AOR=1.33,95% CI: 1.09,1.61). 

 In addition, age, being a widow/divorced and BMI increased the odds of having cardiac vascular disease. By controlling the effect of other variables, the aged 55-64 people were more likely to be at risk of cardiac vascular disease compared to the aged 35-44 people (AOR=1.58, 95% CI: 1.17,2.13). Also, the odds of having cardiac vascular disease was 4.42 times higher for widowed/divorced people than the single ones (AOR=4.42, 95% CI: 1.33,14.72). Furthermore, the odds of having the cardiac vascular disease was higher for people who had BMI≥25 than people who had 18.5≤BMI≤ 24.9. 

We also added the following more findings to the table 6.

Domain Variable Men Women P-value

Self-reported

disease Fracture 106(2.6%) 141(3.5%) 0.947

 Osteoporosis 20(0.5%) 262(6.5) <0.001

We appreciate the reviewer's comments, however, we do believe that the respected reviewer agrees with us that at present, the ongoing study owns enough findings and has generated adequate data, so as, it has merit and could get published as a "Cohort profile". We really hope that you will be considerate to it and our explanations satisfy you. 

In this study, in order to improve the external validity and to increase the generalizability of the study results to the target community, we selected eligible individuals from both sex groups with diverse education and jobs as well as unlike socio-economic status to represent that the designated individuals to be the appropriate representative from the reference population. 

To control and monitor carefully the quality and accuracy of measuring devices such as scales and blood pressure, we calibrated all the concerned devices periodically according to the standard protocol. Furthermore, training and certifying the observers' skills and standardizing the measurement methods were the two other strategies to control information bias that we used to regularly carry out and practice these approaches in our ongoing cohort study. 

To the best of my knowledge, one practical method to identify confounders is to analyze data with or without potential confounders. If the estimation of the association among the independent variables and the outcomes varies considerably by controlling a third variable, then the third variable would be confounding. 

Therefore, in the present study, using statistical methods such as regression models, the relationship between the independent variable and the desired outcome in different categories of confounding variables was calculated separately or controlled by statistical methods such as analysis stratification.

We thank the reviewers and editorial team for helping us improve our manuscript.

We are highly hopeful that the modifications in the manuscript and our explanations satisfy the reviewer. Kindly note we have now reshaped the manuscript by adding and deleting text where most appropriate as per the reviewer's suggestions. 

We really hope that you will be considerate to it.

We are not averse to further modifications of the language or the text. 

We are highly hopeful that the ms finds potential, could fulfill the high standards of the journal and gets positive review as well as acceptance.

Warm regards,

Azim Nejatizadeh, MD., PhD.

Corresponding author

Hormozgan University of Medical Sciences,

Bandar Abbas, Iran

Mobile: +98-9179564291

E-mail: azimnejate@yahoo.com

---

## [Decision Letter · Decision Letter 2]

11 Jul 2021

PONE-D-20-32677R2

Cohort profile: The Bandar Kong prospective cohort study

PLOS ONE

Dear Dr. Nejatizadeh

Thank you for submitting your manuscript to PLOS ONE. After careful consideration, we feel that it has merit but does not fully meet PLOS ONE’s publication criteria as it currently stands. Therefore, we invite you to submit a revised version of the manuscript that addresses the points raised during the review process.

Please see the statistical revision that the reviewer requested you to revisit. This is the last time we can consider the manuscript to be re-reviewed. Please make sure you respond to the minor revision that the reviewer asks for.

Thanks

We look forward to receiving your revised manuscript.

Kind regards,

Hassan Ashktorab

Academic Editor

PLOS ONE

Journal Requirements:

Additional Editor Comments (if provided):

Please see the statistical revision that the reviewer requested you to revisit. This is the last time we can consider the manuscript to be re-reviewed. Please make sure you respond to the minor revision that the reviewer asks for.

Thanks

Reviewers' comments:

Reviewer's Responses to Questions

**Comments to the Author**

1. If the authors have adequately addressed your comments raised in a previous round of review and you feel that this manuscript is now acceptable for publication, you may indicate that here to bypass the “Comments to the Author” section, enter your conflict of interest statement in the “Confidential to Editor” section, and submit your "Accept" recommendation.

Reviewer #3: (No Response)

2. Is the manuscript technically sound, and do the data support the conclusions?

Reviewer #3: Partly

3. Has the statistical analysis been performed appropriately and rigorously? 

Reviewer #3: I Don't Know

4. Have the authors made all data underlying the findings in their manuscript fully available?

Reviewer #3: No

5. Is the manuscript presented in an intelligible fashion and written in standard English?

Reviewer #3: Yes

6. Review Comments to the Author

Reviewer #3: The statistical analysis is acceptable. However,

1- I have some concerns about sample size calculations, I think the collected sample size is too small and setting and OR of 2 or more is rather big difference to detect. I need to see robust evidence that the calculated sample size over 5 years is supported by statistical methods, even for detecting a change in OR of 2.

2-Data collection section is not clearly explained, the population that the data was collected from seems to be different from "'normal" population. Authors claimed that ''70% of the

participants were affected by at least one NCD.'' this cannot be a normal population.

3- P-values reported without explaining what they are testing, for example ''Over 23% of men and 17.3% of women were Hookah smoker (p-value <0.001)'' what is this p-value?

4- Tables are confusing, not enough or missing explanation in the title or body of the tables; for example table 4 and 6 no units are given for variables. Are they percentage or mean or mixed?

7. PLOS authors have the option to publish the peer review history of their article (what does this mean?). If published, this will include your full peer review and any attached files.

Reviewer #3: No

---

## [Author Response · Author response to Decision Letter 2]

31 Jul 2021

The Editor-in-chief 

PLOS ONE

With respect,

Responses to the referee's comments:

Manuscript Number: PONE-D-20-32677

Title: Reply to Manuscript Number: PONE-D-20-32677Title: Cohort Profile: The Bandar Kong Prospective Cohort Study

Article Type: Original research paper (Cohort profile)

Thanks a lot for your critical and peer review of our manuscript (ms). Our manuscript has been accepted with revision. We have provided a point-by-point response to all comments from the reviewers and have revised the manuscript accordingly. 

Therefore, we resubmit our manuscript (ms) titled “Cohort Profile: The Bandar Kong Prospective Cohort Study”, authored by Azim Nejatizadeh et al. Efforts have been made to keep the text to minimum with inclusion of the most relevant references. We edited the text and made the corrections and needful modifications as per the reviewer comments. Notably, we have incorporated the all raised corrections in the revised version. 

Answers to each point raised by the academic editor and reviewer(s)

Journal Requirements:

Responses to the Journal Requirements:

Thanks for the reminder. We went through the reference list and found it comprehensive, however, reference No. 30 was in duplicate the same as reference No. 11. We deleted reference No.30 and kept reference No.11 in the list. The rest of the reference list was accurate and none of the cited papers were retracted. 

Additional Editor Comments (if provided):

Please see the statistical revision that the reviewer requested you to revisit. This is the last time we can consider the manuscript to be re-reviewed. Please make sure you respond to the minor revision that the reviewer asks for.

Reply:

We have provided a point-by-point response to all comments from the reviewers and have revised the manuscript accordingly. We edited the text and made the corrections and needful modifications as per the reviewer comments. Notably, we have incorporated the all raised corrections in the revised version. 

Reviewers'comments:

Reviewer #3: 

The statistical analysis is acceptable. However, 

Comment 1: I have some concerns about sample size calculations, I think the collected sample size is too small and setting and OR of 2 or more is rather big difference to detect. I need to see robust evidence that the calculated sample size over 5 years is supported by statistical methods, even for detecting a change in OR of 2.

Reply 1: About the appropriate sample size, the required sample size was calculated by Fleiss with continuity correction formula, with a two-sided significance level(1-alpha) = 0.95, power (1-beta, % chance of detecting) = 85%, the ratio of sample size of unexposed/exposed = 1, percent of unexposed with outcome = 5%, and risk/ prevalence ratio or odds ratio =1.5. So, it was estimated 2017 people for the exposed group, 2017 non-exposed group, and hence total sample size is 4034 people [Reference].

Reference:

Dean AG, Sullivan KM, Soe MM. OpenEpi: Open Source Epidemiologic Statistics for Public Health, Version. www.OpenEpi.com, updated 2013/04/ 06, accessed 2017/05/06.

Comment 2: Data collection section is not clearly explained, the population that the data was collected from seems to be different from "'normal" population. Authors claimed that ''70% of the participants were affected by at least one NCD.'' this cannot be a normal population.

Reply 2: Thanks for your comment. We did our best to keep the manuscript as short as possible, however, in this particular study, we have to maintain the concept and essence of the study according to the objectives. However, according to your valuable suggestion, we now explained in detail and clearly "Data collection section". Of note, we revisit and checked out the data analysis carefully and found that you are right. The claim ''70% of the participants were affected by at least one NCD", was a rough incorrect estimation and expressed by mistake; hence, we apologize for that. Subsequently, the prevalence of self-reported health conditions of study participants and the most common chronic NCDs diagnosed during the first annual follow-up were listed in Tables 6 and 7 in detail. Thereby, these tabulated data, here, are our reference and quite valid. We really hope that you will be considerate to it and our explanations satisfy you. 

Comment 3: P-values reported without explaining what they are testing, for example ''Over 23% of men and 17.3% of women were Hookah smoker (p-value <0.001)'' what is this p-value?

Reply 3: Thanks for your valuable comment. Below the tables, it was explained which tests were used, as following: 

1 The independent samples t-test was used to determine if there is a significant difference between the means of two groups, the chi-squared test was used to determine whether there is an association between 2 categorical variables.

Comment 4: Tables are confusing, not enough or missing explanation in the title or body of the tables; for example, table 4 and 6 no units are given for variables. Are they percentage or mean or mixed?

Reply 4: Thanks for your valuable comment; you are correct. The title of the tables was modified and units were added within the Tables. Where appropriate, missed percentage symbols were incorporated within the Table 6.

Regarding conclusion section: 

We, again, did our best to draw and support appropriate conclusion based on the findings to date. Now, we included more data and inferred proper conclusions that have been argued in conclusion section. Here, we stressed that prevention of premature deaths due to NCD and decrease of associated health care expenses will be the main objectives of health policy. The consequences of such investigations would aid policy makers to plan and implement profitable facility within the situation of primary health care in Iran, serving as an instance for neighboring countries too.

We also like to keep the manuscript as short as possible, however, in this particular study, we have to maintain the concept and essence of the study. Kindly note we have now revised/modified and reshaped the "conclusion as well as strengths and limitations" sections where most appropriate as per the reviewers’ valuable suggestions. We really hope that you will be considerate to it and our explanations satisfy you. 

Upon acceptance of the manuscript, the raw data will be made available at this site and the access would be free for all academic users under a click wrap protection.

We thank the reviewers and editorial team for helping us improve our manuscript.

We are highly hopeful that the modifications in the manuscript and our explanations satisfy the referees. Kindly note we have now reshaped the manuscript by adding and deleting text where most appropriate as per the reviewers’ valuable suggestions. 

However, we are afraid now with so many of your suggestions that have been incorporated in the text, it has shortened largely, although we revised and tried our best to shorten it much more. We really hope that you will be considerate to it.

We are not averse to further modifications of the language or the text. 

We are highly hopeful that the ms finds potential, could fulfill the high standards of the journal and gets positive review.

Warm regards,

Azim Nejatizadeh, MD., PhD.

Corresponding author

Hormozgan University of Medical Sciences,

Bandar Abbas, Iran

Mobile: +98-9179564291

E-mail: azimnejate@yahoo.com

---

## [Decision Letter · Decision Letter 3]

5 Oct 2021

PONE-D-20-32677R3

Cohort profile: The Bandar Kong prospective cohort study

PLOS ONE

Dear Dr. Nejatizadeh,

Thank you for submitting your manuscript to PLOS ONE. After careful consideration, we feel that it has merit but does not fully meet PLOS ONE’s publication criteria as it currently stands. Therefore, we invite you to submit a revised version of the manuscript that addresses the points raised during the review process.

Please check the numbers in the statistical revised section and correct it based on the reviewer comments. In addition, please have the manuscript re-reviewed for English and typos.

We look forward to receiving your revised manuscript.

Kind regards,

Hassan Ashktorab

Academic Editor

PLOS ONE

Journal Requirements:

Reviewers' comments:

Reviewer's Responses to Questions

**Comments to the Author**

1. If the authors have adequately addressed your comments raised in a previous round of review and you feel that this manuscript is now acceptable for publication, you may indicate that here to bypass the “Comments to the Author” section, enter your conflict of interest statement in the “Confidential to Editor” section, and submit your "Accept" recommendation.

Reviewer #3: (No Response)

2. Is the manuscript technically sound, and do the data support the conclusions?

Reviewer #3: Partly

3. Has the statistical analysis been performed appropriately and rigorously? 

Reviewer #3: No

4. Have the authors made all data underlying the findings in their manuscript fully available?

Reviewer #3: No

5. Is the manuscript presented in an intelligible fashion and written in standard English?

Reviewer #3: Yes

6. Review Comments to the Author

Reviewer #3: The manuscript still needs some corrections and editing. The sample size explained to my previous comment looks OK but it is very different from what is explained under "Study Population" in line 136. Some tables are out of format (table 4 and table 8) and unreadable.

The NCD should be somehow in the title of this study.

SPSS is not any more Statistical Package for Social Science, please correct it.

line 246 to 249 is not technically correct, rewrite it.

You need to edit this manuscript carefully if you want to get it published in PLOS ONE.

7. PLOS authors have the option to publish the peer review history of their article (what does this mean?). If published, this will include your full peer review and any attached files.

Reviewer #3: No

---

## [Author Response · Author response to Decision Letter 3]

18 Nov 2021

The Editor-in-chief 

PLOS ONE

With respect,

Responses to the referee's comments:

Manuscript Number: PONE-D-20-32677R3

Title: Reply to Manuscript Number: PONE-D-20-32677R3

Title: Cohort Profile: Cohort profile: Bandar Kong Prospective study of Chronic Non-Communicable Diseases 

Article Type: Original research paper (Cohort profile)

Thanks a lot for your critical and peer review of our manuscript (ms). Our manuscript has been accepted with minor revision. We have provided a point-by-point response to all comments from the reviewers and have revised the manuscript accordingly. 

Therefore, we resubmit our manuscript (ms) titled “Cohort profile: Bandar Kong Prospective study of Chronic Non-Communicable Diseases ”, authored by Azim Nejatizadeh et al. Efforts have been made to keep the text to minimum with inclusion of the most relevant references. We edited the text and made the corrections and needful modifications as per the reviewer comments. Notably, we have incorporated the all raised corrections in the revised version. 

Answers to each point raised by the academic editor and reviewer(s)

Journal Requirements:

Responses to the Journal Requirements:

Thanks for the reminder. We have not cited papers that have been retracted. while undertaking corrections and modification in the text, to improve the quality of the evidence of the manuscript, we decided to include eight new references to the reference list which have been embedded within the text. These are references No. 2, 5, 27, 33-35, 37, 38. 

Review Comments to the Author

Reviewer #3: 

Comment 1: The manuscript still needs some corrections and editing.

Reply 1: Thanks for your valuable comment; you are correct. We are highly hopeful that the modifications in the manuscript and our explanations satisfy you. Kindly note we have now reshaped the manuscript by adding and deleting text where most appropriate as per your valuable suggestions. We edited and corrected the text thoroughly, however, in this particular study, we have to maintain the concept and essence of the study.

Comment 2: The sample size explained to my previous comment looks OK but it is very different from what is explained under "Study Population" in line 136.

Reply 2: Thanks for your valuable comment. It is modified. The modifications were highlighted in green from line 229 to 233.

Comment 3: Some tables are out of format (table 4 and table 8) and unreadable.

Reply 3: Thanks for your comment. You are correct. We corrected format of the tables 4 and 8 and now are readable. Remaining tables also were organized appropriately.

Comment 4: The NCD should be somehow in the title of this study.

Reply 4: Thanks for your valuable comment. As per your suggestion, we incorporated NCD in the title of the manuscript. Now, it seems much better. 

Comment 5: SPSS is not any more Statistical Package for Social Science, please correct it.

Reply 5: Thanks for your comment. We corrected and modified it.

Comment 6: line 246 to 249 is not technically correct, rewrite it

Reply 6: Thanks for your comment. We have revised and modified it.

Comment 7: You need to edit this manuscript carefully if you want to get it published in PLOS ONE.

Reply 7: Thanks for your comment. Now and again, we further did our best to edit the text carefully and keep the manuscript as short as possible, however, in this particular study, we have to maintain the concept and essence of the study according to the objectives. Therefore, we modified and edited the text wisely and whole manuscript thoroughly as per your suggestion. We really hope that you will be considerate to it where the edited text and our explanations satisfy you. 

Upon acceptance of the manuscript, the raw data will be made available at this site and the access would be free for all academic users under a click wrap protection.

We thank the reviewers and editorial team for helping us improve our manuscript.

We are highly hopeful that the modifications in the manuscript and our explanations satisfy the referees. Kindly note we have now reshaped the manuscript by adding and deleting text where most appropriate as per the reviewers’ valuable suggestions. We really hope that you will be considerate to it.

We are not averse to further modifications of the language or the text. 

We are highly hopeful that the ms finds potential, could fulfill the high standards of the journal and gets positive review.

Warm regards,

Azim Nejatizadeh, MD., PhD.

Corresponding author

Hormozgan University of Medical Sciences,

Bandar Abbas, Iran

Mobile: +98-9179564291

E-mail: azimnejate@yahoo.com

---

## [Decision Letter · Decision Letter 4]

2 Mar 2022

Cohort profile: Bandar Kong Prospective study of Chronic Non-Communicable Diseases

PONE-D-20-32677R4

Dear Dr. Nejatizadeh,

We’re pleased to inform you that your manuscript has been judged scientifically suitable for publication and will be formally accepted for publication once it meets all outstanding technical requirements.

Kind regards,

Rajnish Joshi

Academic Editor

PLOS ONE

Additional Editor Comments (optional):

Reviewers' comments:

Reviewer's Responses to Questions

**Comments to the Author**

1. If the authors have adequately addressed your comments raised in a previous round of review and you feel that this manuscript is now acceptable for publication, you may indicate that here to bypass the “Comments to the Author” section, enter your conflict of interest statement in the “Confidential to Editor” section, and submit your "Accept" recommendation.

Reviewer #3: All comments have been addressed

2. Is the manuscript technically sound, and do the data support the conclusions?

Reviewer #3: Yes

3. Has the statistical analysis been performed appropriately and rigorously? 

Reviewer #3: Yes

4. Have the authors made all data underlying the findings in their manuscript fully available?

Reviewer #3: Yes

5. Is the manuscript presented in an intelligible fashion and written in standard English?

Reviewer #3: Yes

6. Review Comments to the Author

Reviewer #3: All comments are appropriately addressed, wish you the best in your research an in your job. Looking forward for more quality reserch output.

7. PLOS authors have the option to publish the peer review history of their article (what does this mean?). If published, this will include your full peer review and any attached files.

Reviewer #3: No

---

## [Editor Report · Acceptance letter]

14 Mar 2022

PONE-D-20-32677R4 

Cohort profile: Bandar Kong Prospective study of Chronic Non-Communicable Diseases  

Dear Dr. Nejatizadeh:

I'm pleased to inform you that your manuscript has been deemed suitable for publication in PLOS ONE. Congratulations! Your manuscript is now with our production department. 

Kind regards, 

on behalf of

Dr. Rajnish Joshi 

Academic Editor

PLOS ONE